# Diverging hydrological drought traits over Europe with global warming

Carmelo Cammalleri[*], Gustavo Naumann, Lorenzo Mentaschi, Bernard Bisselink, Emiliano Gelati,

Ad De Roo and Luc Feyen

European Commission, Joint Research Centre (JRC), 21027 Ispra (VA), Italy.

[*] Correspondence: carmelo.cammalleri@ec.europa.eu; Tel.: +39-0332-78-9869.

**Abstract**

Climate change is anticipated to alter the demand and supply of water at the earth's surface. Since many societal impacts from a lack of water happen under drought conditions, it is important to understand how droughts may develop with climate change. This study shows how hydrological droughts will change across Europe with increasing global warming levels (GWL of 1.5, 2 and 3 K above preindustrial temperature). We employed a low-flow analysis based on river discharge simulations of the LISFLOOD spatially-distributed physically-based hydrological and water use model, which was forced with a large ensemble of regional climate model projections under a high emissions (RCP8.5) and moderate mitigation (RCP4.5) pathway. Different traits of drought, including severity, duration and frequency, were investigated using the threshold level method. The projected changes in these traits identify four main sub-regions in Europe that are characterized by somehow homogeneous and distinct behaviours with a clear southwest/northeast contrast. The Mediterranean and Boreal sub-regions of Europe show strong, but opposite, changes at all three GWLs, with the former area mostly characterized by stronger droughts (with larger differences at 3 K) while the latter is expected to experience a reduction in all drought traits. In the Atlantic and Continental sub-regions the changes are expected to be less marked and characterized by a larger uncertainty, especially at the 1.5 and 2 K GWLs. Combining the projections in drought hazard with population and agricultural information shows that with 3 K global warming an additional 11

million people and 4.5 million ha of agricultural land are projected to be exposed to droughts every
year, on average, with the most affected areas located in the Mediterranean and Atlantic regions of
Europe.
**Keywords:** climate change, LISFLOOD, drought, low-flow analysis, Paris agreement, global
warming levels, human water use

## 1. Introduction

As a natural phenomenon, drought occurs in all climates due to a temporary lack of precipitation, which can propagate through the different compartments of the water cycle (Van Loon and Van Lanen, 2012). Drought conditions can be exacerbated by high temperatures, causing an increase in evapotranspiration demand and soil water content draining (e.g., Teuling et al., 2013), and their impacts can be further intensified in areas with an overexploitation of available water resources (Van Loon and Van Lanen, 2013). The strong dependency of drought conditions on the key meteorological forcing suggests likely effects of climate change on future drought severity, duration and frequency, mainly through an alteration of the water balance dynamics (Stagl et al., 2014).

Depending on the degree of penetration of the water deficit into the hydrological cycle, drought is commonly classified into meteorological (e.g., precipitation), agricultural (e.g., soil moisture) and hydrological (e.g., river discharge) drought (Wilhite, 2000). Each drought type may be perceived most relevant for a specific application, and different indicators may capture different effects of climate change (Feng, 2017). In spite of the strong connection between the socioeconomic impacts of droughts and negative soil moisture and river discharge anomalies, fewer studies (e.g., Samaniego et al., 2018; Forzieri et al., 2014) have focused on the impact of climate change on agricultural and hydrological droughts at European scale compared to meteorological events (e.g., Heinrich and Gobiet, 2012; Spinoni et al., 2018). This focus on meteorological drought mainly relates to the relative simplicity and lower input data requirements of calculating meteorological drought indicators (i.e., Standardised Precipitation Index, SPI) compared to agricultural and hydrological drought indices, whose analysis usually requires simulations from hydrological models, as also highlighted by the larger emphasis placed on meteorological drought hazard in operational monitoring systems (Barker et al., 2016). Scientific and practical interest in hydrological drought is motivated by the direct and indirect impacts on several socioeconomic sectors, such as energy production, inland water transportation (Meyer et al., 2013), irrigated

agriculture, and public water supply (see the European Drought Impact Inventory, https://www.geo.uio.no/edc/droughtdb/), as well as causing losses of ecosystem and biodiversity (Crausbay and Ramirez, 2017). In particular, streamflow drought complements meteorological and soil moisture droughts thanks to its more rapid response to precipitation aberrations compared to groundwater (Tallaksen and van Lanen, 2004).

With the raising awareness of climate change, a number of local and regional studies assessed the potential impacts of climate change on hydrological drought in recent years (e.g., Brunner et al., 2019; Cervi et al., 2018; Hellwig and Stahl, 2018; Nerantzaki et al., 2019; Rudd et al., 2019; Van Tiel et al., 2018). These studies provided highly detailed insights on the local processes, but the limited extent of their spatial domain and lack of homogeneity in the adopted drought indicators, modelling framework and climate scenarios complicated the understanding of large-scale patterns of changes. In spite of the value of continental-scale analyses, few studies have looked at how hydrological droughts could develop across Europe with climate change. They are typically based on pan-European hydrological models forced by climate projections (Feyen and Dankers, 2009; Forzieri et al., 2014; Lehner et al., 2006; Marx et al., 2018; Roudier et al., 2016), with ever improved representation of processes in the hydrological models. These improvements included accounting for the effects of water use, more detail in the climate projections (by the use of higher resolution regional climate models), and better accounting for climate uncertainty through multi-model ensembles.

Most past studies portrayed how drought conditions across Europe could look at future points in time (mid- or end- of century) for alternative scenarios of greenhouse gas emissions. However, following the UNFCCC (United Nations Framework Convention on Climate Change) Paris Agreement (UNFCCC, 2015) and the focus on limiting the increase in global average temperature to well below 2 K above the pre-industrial level, the paradigm in climate change studies has started to shift from analysing the effects at specific future time windows to evaluating the effect at specific global warming levels (GWLs). To date, there are only few studies that provided insights on how

hydrological droughts could change at different GWLs. Roudier et al. (2016) used three hydrological models forced with high resolution regional climate projections to evaluate changes in 10- and 100-year streamflow drought events, with a focus solely on the 2 K scenario. Marx et al. (2018) used three different hydrological models forced by coarse-resolution global climate projections that were downscaled accounting for altitude effects in temperature and precipitation. They used a simple 90-th percentile of exceedance of river discharge as index, which is representative of the low-flow spectrum. Both studies did not consider water consumption, which is key to represent feedbacks between droughts and human activities (Van Loon et al., 2016).

To further deepen the understanding of the influence of climate change and water use on future droughts, the daily streamflow simulations for the pan-European river network obtained with the LISFLOOD spatially-distributed hydrological model, forced with an ensemble of 11 bias-corrected regional climate projections for RCP4.5 and RCP8.5 (Moss et al., 2010), were used. The model incorporates water use modules to reproduce the major sectorial water demands, accounting for the human impact on streamflow propagation, and resulting in a streamflow deficit that represents the integrated deficiency in water availability over the entire upstream catchment.

These streamflow simulations were analysed with the twofold goal of: i) evaluate changes in hydrological droughts across Europe between present climate and climate corresponding to different GWLs, and ii) quantify the effects of the projected changes on two of the main exposed compartments. Specifically, we look at 1.5, 2 and 3 K global warming, which represent the different Paris agreement climate change mitigation targets, and we exploited the threshold level method for event extraction, which allows for a detailed extreme value analysis of different streamflow drought traits, including severity, duration and frequency. The effects of the projected changes on two key exposed quantities is also evaluated through a drought exposure analysis, with a specific focus on the changes between the present and future exposed population and agricultural land, which are representative quantities in the major social and economic sectors impacted by drought in Europe (e.g., agriculture and livestock farming, and public water supply).

## 2. Materials and Methods

### 2.1 Climate forcing

In this study, we used projections from 11 combinations of global and regional climate models under two Representative Concentration Pathways (RCP4.5 and RCP8.5) obtained from the EURO-CORDEX initiative (Jacob et al., 2014). The climate projections used in this study were produced by Dosio (2020) by applying a bias-correction quantile mapping approach (Dosio et al., 2012) using the observational dataset EOBSv10 (Haylock et al., 2008). The analysis focused on 30-year time windows centred on the year when the global models project an increase in global average temperature of 1.5, 2 and 3 K above preindustrial (1881-1910) temperature. For these periods, drought characteristics were contrasted against those derived for the baseline reference period (1981-2010), which has a 0.7 K temperature increase compared to the preindustrial period.

Across all models, the two RCPs reach the 1.5 and 2 K GWLs around the year 2030 and 2053 (RCP4.5), 2025 and 2040 (RCP8.5), on average. The RCP8.5 simulations reach the 3 K GWL in 2063 on average, whereas only one model reaches 3 K warming for RCP4.5. According to the independence of the projected river flow changes from the adopted pathway observed in Mentaschi et al. (2020) for annual minimum (drought), average and maximum (flood) flows, we assumed that a single multi-model ensemble can be obtained by merging the outputs from both RCPs. Given that only one model reaches 3 K warming for RCP4.5, the model ensemble was composed by a total of 22 members for the 1.5 and 2 K GWLs and only 12 members for the 3 K GWL.

### 2.2 Hydrological modelling

Simulations of daily river discharge ($Q$) were produced at a $5 \times 5$ km spatial resolution over Europe by forcing the LISFLOOD model (De Roo, 2000) with the bias-corrected climate projections. LISFLOOD is a spatially-distributed physically-based hydrological model that simulates all the main hydrological processes occurring in the land-atmosphere system, including evapotranspiration fluxes (separately for crop transpiration and direct evaporation), infiltration

(Xinanjiang model), soil water redistribution in the vadose zone (Darcy 1-D vertical flow model),
groundwater dynamics (two parallel linear reservoirs), snow accumulation and melt (degree-day
factor method) and surface runoff (for further details on each module, see Burek et al., 2013). The
surface runoff generated in each cell is channelled to the nearest river network cell by means of a
routing component based on a 4-point implicit finite-difference solution of the kinematic wave
(Chow et al., 1988).
The water abstractions component in LISFLOOD consist of five modules: (manufacturing)
industrial, energy, livestock, domestic and irrigation water demand. While irrigation water demand
is modelled dynamically within LISFLOOD, the other four components are downscaled to the
model grid cells from country-level data obtained from EUROSTAT and AQUASTAT. High
resolution data from the Land-Use based Integrated Sustainability Assessment (LUISA) Territorial
Modelling Platform (Jacobs-Crisioni et al., 2017) were used for the spatial downscaling.
In detail, irrigation was estimated dynamically at the model time step (daily in this study)
based on two distinct methods for crop irrigation and paddy-rice irrigation, as defined from land use
maps. In the former, the demanded water amount by the crop (transpiration) is compared to the
available water in the soil and the irrigation is modelled to keep the soil water content at field
capacity (also accounting for the different efficiency of the irrigation systems). In the paddy-rice
irrigation instead, a defined water-level is maintained during the whole irrigation season (also
accounting for soil percolation).
Livestock water demand at grid scale was modelled as described in Mubareka et al. (2013), by
computing the water demand of each livestock category (e.g., cattle, pigs, sheep) from livestock
density maps and literature water requirements. Public water withdrawal was downscaled to model
resolution using a land use proxy approach (Vandecasteele et al., 2014), assuming that public water
withdrawal is the total water withdrawn in populated areas (i.e., water usage from
commercial/service are negligible). Similarly, industrial water demand was disaggregated using the
industry/commerce land use class in the LUISA platform (Bisselink et al., 2018), Water demand for
energy and cooling was computed with a relatively similar approach, with national data downscaled
to the locations of large power thermal power stations registered in the European Pollutant Release
and Transfer Register data base (E-PRTR).
Future projections of the main socioeconomic drivers of water use are based on the EU
economic, budgetary, and demographic projections (EC, 2015), and the European energy reference
scenario (Capros et al., 2013) available in the LUISA platform. Irrigation demand was modelled
based on projected agriculture land use changes and the dynamic climate-dependent water
requirements. Projections of future industrial water demand were based on the Gross Value Added
of the industrial sector available from the GEM-E3 model (Capros et al., 2013). Future changes in
energy water use were simulated according to the electricity consumption projections from the
POLES model (Prospective Outlook on Long-term Energy Systems, Keramidas et al., 2017). Future
domestic water demand was estimated based on spatially detailed (100 × 100 m) projected
population maps. Due to the absence of information on future livestock in LUISA, the
corresponding water demand was kept constant. Considering the relatively limited extent of area
with high livestock water demand (Mubareka et al., 2013), only small effects are expected due to
this assumption. As the EU projections do not go up to the end of the end of the century, projections
of water use are dynamic only up to 2050 and were kept constant afterwards.
The LISFLOOD modelling framework have been successfully applied in Feyen and Dankers
(2009) and Forzieri et al. (2014) in previous studies on drought future projections. In these analyses,
model simulations were validated against long records (more than 30 years) of streamflow data
from several gauging stations (209 and 446 stations, respectively), obtaining satisfactory results on
quantities such as annual minima and deficit. Gauging stations were mostly located in western and
central Europe, where both studies highlighted less reliable performances during the frost season.
The most recent calibration and validation exercise of LISFLOOD over the European domain
has been performed over more than 700 stations as part of the EFAS (https://www.efas.eu/) flood
early warning systems (Arnal et al., 2019). The calibration procedure is based on the Evolutionary
Algorithm described in Hirpa et al. (2018), and it adopted the Kling-Gupta Efficiency (KGE; Gupta
et al., 2009) as the objective function in order to target an optimization of three quantities: total
volume, the spread of the flow (e.g. flow duration curve), and the timing and shape of the
hydrograph (Yilmaz et al., 2008).

### *2.3 Drought modelling*

The hydrological drought modelling approach used in this study is analogous to the
methodology used to estimate the low-flow indicator developed as part of the European Drought
Observatory (EDO) (Cammalleri et al., 2017). The key quantity is the water deficit computed from
an unbroken sequence of discharge ($Q$) values below a defined low-flow threshold. We used the 85-
th percentile of exceedance, $Q_{85}$, derived for the present climate as a threshold both in the present
and future scenarios, with the aim to estimate how droughts under present climate conditions will be
projected under climate change.
According to the theory of runs (Yevjevich, 1967), a continuous period with river flow values
below the defined low-flow threshold was considered as a drought event, of which the severity was
quantified by the total deficit ($D$, represented by the area enclosed between the threshold and the
streamflow time series). Other key traits of drought derived from the analysis were the duration,
quantified by the length of the drought in days ($N$), and the frequency of the events, which can be
expressed as return period ($T$).
In order to avoid potential bias in the analysis with the inclusion of minor events and to ensure
the independence among events, two post-processing corrections were applied after selection of the
events below the threshold: 1) small isolated events (of duration less than 5 days) were removed
from the analysis (Jakubowski and Radczuk, 2004), and 2) consecutive events with an inter-event
time smaller than 10 days were pooled together (Zelenhasić and Salvai, 1987).
Following this drought definition, a sequence of events for both the baseline period and the
three GWLs was derived. Given the large variability of $D$ values across the European domain due to
differences in hydrological regimes and size of river basins, the changes in drought severity were
expressed as relative differences (%) from the values in the baseline period (1981-2010). The series
of $D$ events was fitted according to the Pareto Type II distribution (also known as Lomax
distribution, a special case of the Generalized Pareto Distribution), formally expressed as (Lomax,

217  1987):

$$F(D;\alpha;\lambda) = 1 - \left(1 + \frac{D}{\lambda}\right)^{\alpha}$$ (1)
where $\alpha$ and $\lambda$ are the strictly positive shape and scale parameters, respectively, derived from the
sample according to the maximum likelihood method. The fitted distributions allowed computing
the return period associated to a specific $D$ value ($T$, the average occurrence interval which refers to
the expected value of the number of realizations to be awaited before observing an event whose
magnitude exceeds $D$; Serinaldi, 2015), or to be used in reverse to estimate the $D$ value associated
to a specific return period.
The same drought modelling approach was previously tested in Cammalleri et al. (2017) and
Cammalleri et al. (2020) for the development of a low-flow indicator as part of the European and
Global Drought Observatories (EDO and GDO, https://edo.jrc.ec.europa.eu). These tests included
assessments for some major past drought events, as well as goodness-of-fit test for the Lomax
distribution for both European and Global river basins. Within EDO and GDO, regular monthly
drought    reports    are    also    produced    in    case    of    significant    drought    events
(https://edo.jrc.ec.europa.eu/edov2/php/index.php?id=1051), which also systematically evaluate the
capability of the low-flow index to capture the dynamic of hydrological droughts.
*2.4  Population and agricultural land exposed to streamflow drought*
In order to quantify how global warming could change exposure to streamflow drought in
Europe, different exposed quantities can be analysed depending on the impacted sector. Among the
15   impact   categories   available   in   the   European   Drought   Impact   Inventory   (EDII,
https://www.geo.uio.no/edc/droughtdb/), agriculture and livestock farming (category 1), and public
water supply (category 7) are the two most reported sectors. As a consequence, we decided to focus
the exposure analysis on population and agricultural land, as quantities strongly related to these two
categories. For the baseline we used the map of agricultural areas from the CORINE land Cover
(EEA, 2016) and the population density from the LUISA Territorial Modelling Platform (Batista e
Silva et al., 2013). Consistently with the water use simulations with socioeconomic dynamics up to
2050, for future exposure the LUISA land use and population projections of 2050 were used.
The spatial data of population and agricultural land were summed over NUTS 2 statistical
regions (or equivalent for EU-neighbour countries according to EUROSTAT,
https://ec.europa.eu/eurostat/web/nuts/statistical-regions-outside-eu). Similarly, the median change
in drought frequency of an event with a 10-year return period in the baseline was computed from all
the cells within a NUTS 2 region. These quantities allowed computing the expected changes in
exposed population and agricultural land, which were then equally divided over the 10-year period
to obtain a standardized year-average quantity. Finally, changes over NUTS 2 regions were further
aggregated to country scale.
**3. Results**
*3.1 Evaluation of the changes in main drought traits*
*3.1.1 Drought severity*
Figure 1 shows the ensemble-median relative change in severity of a 10-year drought between
the baseline and the GWLs, with positive (negative) values indicating a higher (lower) drought
severity with warming compared to the reference. In order to assess the robustness of the ensemble
median values, the projected changes are considered robust only if at least 2/3 of the ensemble
members agree on the sign of change (no-agreement otherwise), which is a simplification of the
approach proposed by Tebaldi et al. (2011) and applied over Europe by Dosio and Fischer (2018).
The spatial maps depicted in Figure 1 highlight a strong divergence in the projected changes of
drought severity with warming over Europe, with four macro-regions (delimited in Figure 1 lower-
right panel) displaying somewhat homogeneous behaviour. The four macro-regions were derived by
computing for each country the predominant change for the three GWLs, then by combining the
countries with similar features. These macro-regions are in line with the ones defined in the IPCC
AR5 subdivision for Europe (Kovats et al., 2014; Metzeger et al., 2005), and they have been already
used in previous early studies at continental-scale (i.e., Feyen and Dankers, 2009; Lehner et al.,
2006). These four macro-regions are adopted in all the subsequent analyses.
In the Mediterranean sub-region (i.e., Iberian Peninsula, Italy, Greece and the Balkans)
generally more severe droughts are projected, whereas in the Boreal sub-area (i.e., Scandinavia
peninsula and Baltic countries) drought severity is expected to reduce almost everywhere. The
projected changes are less marked in two transition regions, but, in general, they point towards more
severe droughts in the Atlantic sub-region (i.e., British Isles, France, Belgium and the Netherlands)
and less severe droughts over the Continental sub-area (Germany, Poland and eastern European
countries). Overall, these patterns of change become stronger and more robust with increasing
warming.
The strongest increase in drought severity is projected for Portugal, Spain and Greece, where
the fraction of rivers with an increase in deficit of more than 50% at 3 K is 99, 80 and 75%,
respectively. If climate stabilizes at 2 K, streamflow drought severity is lower than at 3 K, but still
at least 50% higher than in the baseline for half of the rivers of Portugal and Spain, and 35% of
Greece. Capping global warming at 1.5 K would further limit the increase in severity, with only 21,
20 and 14% of the rivers of Portugal, Spain and Greece expected to experience an increase in
drought severity of more than 50%.
Over the Atlantic region (apart from Iceland), streamflow droughts are generally projected to
also become more severe with global warming. The south of France shows a pattern towards more
severe flow deficits with warming that is similar to that projected for most of the Mediterranean.
For the other parts of the Atlantic sub-region the changes are less pronounced. Keeping warming to
2 K or below would limit the increase in severity for most of the region to below 25% compared to
the baseline. At 3 K warming, the increase in severity could reach up to 50%. In some parts of the
Atlantic sub-region, such as the Seine river catchment in France (northern France), at lower levels
of warming the climate models do not agree on the sign of the change, or show a small trend
towards less severe droughts. Yet, with stronger warming the signal of change reverses towards
more severe droughts.
Over most of the Continental sub-region there is a trend towards less severe droughts with
global warming. On the one hand, this trend is somewhat more pronounced in upstream Danube
tributaries that drain the Alps to the east. In many downstream Danube tributaries in Hungary,
Romania and Bulgaria, on the other hand, streamflow droughts are projected to become more severe
(in agreement with the results reported in Stagl and Hattermann, 2015). At low levels of global
warming (1.5 and 2 K) most of Germany is expected to experience less severe droughts. At high
levels of warming (3 K), however, western parts of Germany are projected to experience and
inverse trend while the rest of the region shows a large uncertainty in the projected changes. In
contrast to most of the Continental sub-area, projections of streamflow drought severity show an
increase with global warming over the main rivers in Denmark.
Finally, in most of the Boreal region, streamflow drought deficits is expected to become
progressively less severe with warming. At 3 K warming streamflow droughts could be half as
severe compared to the baseline, with few notable exceptions in southern Sweden.
*3.1.2  Drought duration*
Figure 2 shows the fraction of each sub-region (presented in the lower-right panel of Figure 1)
for which a certain degree of change in drought duration (compared to the reference period) is
projected for the different warming levels. There is a clear upward climate change-induced trend in
the fraction of the Mediterranean sub-region that will be exposed to longer droughts with increasing
GWL. When keeping global warming limited to 1.5 K, droughts are projected to last more than 5-
days longer in about 40% of the Mediterranean, with a prolongation above 15 days in slightly more
than 5% of the area. At 3 K warming, however, streamflow droughts will last longer than in the
reference period in 80% of the area and nearly half of the sub-region could face an increase in
drought duration of at least 10 days.
An upward, but less pronounced, trend in drought duration with global warming is also
projected for most of the Atlantic sub-region. At 1.5 K GWL, the area with negative changes in
drought duration (about 30%) is comparable to the area with positive changes, with no clear signal
in about 40% of the domain. With higher levels of warming, the area with a shorter drought
duration compared to the reference shrinks, while the fraction of land that is expected to face longer
droughts steadily expands. Compared to 1981-2010, droughts are projected to last longer in about
75% of the sub-region at 3 K GWL, hence similar to what can be observed for the Mediterranean.
Yet, for only 10% of the area, drought duration is expected to increase by more than 10 days.
In the Continental sub-region, the area that shows a decrease in drought duration compared to
the reference period is around 65% at 1.5 K, which slightly reduces in extent with increasing
warming. Yet, over this area droughts are expected to progressively shorten with further warming.
At 3 K warming, with positive changes of at least 10 and 15 days over more than 30 and 10% of the
region, respectively. Drought duration is projected to increase over a small part (20% at 3 K) of the
domain compared to the reference period, mainly corresponding to Bulgaria.
Over the Boreal sub-region, droughts are projected to become shorter with global warming over
practically the whole domain. At 1.5 K warming, drought duration is expected to be at least 15 days
shorter than in 1981-2010 in 20% of the area, which grows to 50% of the area at 3 K warming. For
all sub-regions, the fraction of area with no-agreement in future drought duration changes tends to
reduce with increasing global warming, and this signal is very consistent among all the climate
projections. At 3 K warming, projections show that less than 15% of the domain under study have
no agreement in the direction of change in drought duration.

### 3.1.3   Drought frequency

Figure 3 shows the frequency density of drought return periods for the three GWLs
corresponding to an event with a return period ($T$) of 10 years under baseline climate. In these plots,
values greater than 10 can be interpreted as a reduction in drought frequency (an event with $T = 10$
years in the baseline will become rarer), whereas values lower than 10 represent an increase in
drought frequency (an event with $T = 10$ years in the baseline will become more common).
The frequency distributions of $T$ values for the Mediterranean (upper-left panel) show a clear
shift towards more recurrent droughts. At 1.5 K warming the peak value is around 8 years, which
further reduces to 7 and 6 years at 2 and 3 K warming, respectively. At 3 K warming the lower tail
of the distribution falls below 4 years. In nearly 10% of the rivers, drought deficits that in baseline
climate happen once in 10 years are expected to occur at least 2.5 times more frequent with 3 K
warming. In the Atlantic sub-region the central value also reduces with warming, yet the overall
reduction is less pronounced than in the Mediterranean sub-area, with a median value around 7
years at 3 K warming. In the Continental region, droughts will in general become less frequent with
a central value between 12 and 13 years at all warming levels, even if the fraction of river cells with
an increase in frequency (around 28% at 3 K) is larger than that with an increase in drought duration
(less than 20% at 3 K, see Figure 2). In the Boreal sub-area the shift towards less frequent droughts
is much more pronounced, with projected return periods concentrated around 20, 30 and 40 years
for 1.5, 2 and 3 K warming, respectively.
Changes in the frequency density plots can be observed not only in the central tendency values,
but also in the spread, which increases with warming for all regions. Additionally, changes opposite
to the general trend can be observed in all regions. For example, over very few locations in the
Mediterranean sub-region, such as some Alpine mountain drainage basins in northern Italy, drought

conditions could become less severe and frequent (see also drought severity changes in Figure 1). In the Atlantic region, the small secondary peak of *T* values > 20 years corresponds to areas where droughts are projected to occur less frequently with global warming, such as Iceland and few tributaries from the Rhône that originate in the Alps (similarly to what was observed on drought severity in Figure 1). Even in the Boreal region a small fraction of the sub-domain shows an increase in drought frequency, while drought duration is projected to reduce practically everywhere. Over this region, the presence of small areas with increase in frequency causes a slight reduction in the frequency median value at 3 K GWL (26 years, compared to 27 years at 2 K) even if the peak shifts to the right with warming (i.e. less frequent droughts).

The results reported in Figure 3 for the 10-year return period can be seen as representative of the behaviour at other return periods as well. To support this consideration, the data in Figure 4 report the sub-region median relative changes at the three GWLs for events with a baseline return period of 3, 5, 10, 20 and 50 years. The plots clearly show how all the return periods have similar dynamics, with the only notable exception represented by the more marked reduction in median relative change of high return periods for the 3 K GWL in the Boreal sub-region (i.e., 20 and 50 years). It is also worth to point out how even if the dynamics are comparable among the different return periods, the magnitude of the relative changes is higher for the longer return periods (i.e. the rarer events).

### *3.2 Population and agricultural land exposed to drought*

Figure 5 shows the changes with respect to the baseline in population projected to be exposed to streamflow drought at country scale (percentage relative changes are also reported as numbers next to the bars). Total changes for the four macro-regions and the entire domain (TOTAL) are summarised in Table 1. Aggregated over the whole domain, about 1.5 million fewer people are expected to be annually exposed to drought at 1.5 K GWL compared to the baseline period, which reverses to an increase of about 2.5 and 11 million people/year compared to baseline human

exposure at 2 and 3 K GWLs, respectively. This shift in the sign of the changes is caused by the fact that at 1.5 K the increase in population exposed annually in the Mediterranean (2.4 million) and Atlantic (less than 0.1 million) sub-regions is outweighed by the reduction in exposure in the Boreal (-0.6 million) and, most importantly, Continental (-3.4 million) sub-regions. Projections in the Mediterranean and Atlantic sub-regions show a progressive increase in population exposed (up to a total of 15.8 million people/year for 3 K GWL over the two regions), while in the Boreal and Continental combined human exposure to droughts is expected to remain roughly the same for all three GWLs (i.e., -3.9, -5.4 and -4.7 million/year at 1.5, 2 and 3 K, respectively).

Spain is projected to have the largest absolute increase in population exposed to drought with global warming, with an almost doubling (+3.8 million/year) of the number of people exposed to drought each year at 3 K GWL. In relative terms, the relative increase in population exposure at 3K is also high in Portugal (+81%), United Kingdom (+58%) and France (+52%). The largest absolute decrease in population exposed is expected for Germany at 1.5 and 2 K GWL (-1.8 and -1.7 million people/year) and Poland at 3 K GWL. The transition of several areas in Germany from a decrease in drought to uncertain conditions (see as an example western Germany in Figure 1) explains the lower number of exposed people at 3 K (-0.9 million people/year) compared to Poland (-1.2 million people/year). The strongest reduction in population exposure in relative terms is expected for Norway, Iceland and Lithuania (up to 65, 87 and 85%, respectively).

Exposure of agricultural land (Figure 6 and Table 2) shows similar trends as for population. Aggregated over Europe, the change in exposure is projected to be balanced in the exposed agricultural land at 1.5 K GWL (net increase of 0.1 million ha/year), whereas at higher warming levels exposure of agricultural land increases to 1.2 and 4.5 million ha/year at 2 and 3 K, respectively. This increasing trend in the Europe-average changes can be explained by the expected steady increase in agricultural land exposed to drought in the Mediterranean and Atlantic sub-regions (up to 6 million ha/year combined at 3 K), which is not counterbalanced at the highest warming by the agricultural land being less exposed to drought in the Boreal and the Continental

sub-regions (-1.3 million ha/year at 1.5 K and -1.5 million ha/year at 3 K). In absolute numbers,
Spain shows the largest projected increase in the agricultural land exposed at all GWLs, with an
additional 0.9 million ha/year at 1.5 K to 2.6 million ha/year at 3 K (corresponding to a relative
increase of about 35 and 97%, respectively). Relative changes are expected to be quite notable for
other Mediterranean countries as well, such as Portugal and Greece, reaching almost 120 and 77%
at 3 K, respectively.
**4.  Discussion**
The projections of severity, duration and frequency underline some common features in future
streamflow drought in Europe. The uncertainty in the projections is more marked at the 1.5 and 2 K
GWLs, whereas change patterns are more statistically robust at higher warming, as also observed by
Marx et al. (2018) for minimum flows. Overall, the magnitude of the projected changes increases
with warming for all the drought traits, with only limited areas interested by an inversion in the
trend. The main pattern is a strengthening of the dichotomy between south-western and north-
eastern Europe, with the already drought-prone south-west becoming even more prone to droughts
while the north-east will experience a further wetting. This result suggests a continuation of a trend
that is already ongoing according to Stagge et al. (2017), and it is also in line with other studies that
projected streamflow droughts focusing on specific time periods instead of GWLs (Lehner et al.,
2006; Feyen and Dankers, 2009; Stahl et al., 2012; Forzieri et al., 2014) or on agricultural (e.g.,
Samaniego et al., 2018) and meteorological (e.g., Gudmundsson and Seneviratne, 2016; Spinoni et
al., 2018) droughts. Hence, there is growing consensus in the community on the main patterns of
climate-induced changes on drought conditions in Europe.
Overall, the Mediterranean sub-region shows the strongest increase in drought traits, with
droughts projected to become more severe, last longer and happen more frequently already at 1.5 K
GWL. The combined effects of increasing temperature and decreasing summer precipitation
(Dubrovský et al., 2014; Vautard et al., 2014) are expected to result in a further exacerbation of

water deficits in an area already prone to limited water resources. This is particularly true during summer, because of high water abstraction for irrigation (about 60% of the current water demand, Vandecasteele et al., 2014). Studies that present future scenarios in agricultural water demand (i.e. Chaturvedi et al., 2015; Schmitz et al., 2013) suggest that improvements in irrigation efficiency could mitigate these impacts. Overall, the increasing pressure of drought on this region agrees with global studies that identify the Mediterranean as a hot spot for climate change, even if the targets set by the Paris agreement will be met (Gu et al., 2020), and also with the study of Guerreiro et al. (2017) on the potential occurrence of multi-year droughts in major Iberian water resource regions.

In contrast, the Boreal sub-region is projected to experience a general reduction in all drought traits, as the increase in precipitation will likely outweigh the increase in evaporative demand due to elevated temperatures (Jacob et al., 2018). Over this region, similarly to the Alps (Donnelly et al., 2017), increasing winter precipitation and higher temperatures are expected to result in higher winter flows, when river flows are typically at their lowest (Gobiet et al., 2014). This result is obtained in spite of the projected general increase in public water demand (the highest share of total withdraws in northern Europe) and business-as-usual per capita water use (Vandecasteele et al., 2014).

In the other two sub-regions the projections are less uniform, with more variation in the signal and robustness of the projections with global warming. In the Atlantic sub-region the increase in droughts at 3 K is expected to be less pronounced compared to the Mediterranean, but similarly robust, while at lower warming levels there is large uncertainty in the projections. In some river basins, such as the Seine in northern France, a decrease in droughts or uncertain trend is projected for low levels of global warming, while at higher levels of warming drought conditions are projected to worsen. This shift in the sign of the changes is likely related to the fact that at higher levels of warming the atmospheric demand (evapotranspiration) rises faster than supply (precipitation) due to the combination of a strong rise in temperature and a slight or uncertain increase in annual precipitation and a decline in summer precipitation (Kotlarski et al., 2014). In the

Atlantic sub-region, areas with projected strong increase in population (e.g. southern UK, EUROSTAT, 2019), are the ones with a clear increase in droughts for all warming levels. Given the role of population in domestic water demand, changes over these regions seems to further exacerbate the climate effects.

In the Continental sub-region the projected overall decrease in droughts is rather inhomogeneous in strength. In upstream Danube tributaries draining the Alps there is a strong trend towards less severe droughts as winter flows increase due to changes in snow accumulation and melt caused by increased winter precipitation and higher temperatures (Forzieri et al., 2014; Marx et al., 2018). In downstream reaches of the Danube, more severe droughts are projected due to a reduction in summer flows caused by an increased evaporative demand and less precipitation, as well as the reduced snowmelt contribution from the Alps (Jenicek et al., 2018). Also, in Germany, the trend towards less severe droughts is reversed at higher warming as the increasing natural and human demand in drier summers outbalance higher annual supply. The revert to increase in droughts at 3 K GWL is the case especially in western parts of Germany such as downstream reaches of the Rhine (Bosshard and Kotlarski, 2014).

The heterogeneity in the strength of the outcomes obtained over the Continental sub-region further stress how the complex interplay between supply (precipitation), atmospheric demand (evapotranspiration) and human water use can result in different projected trends. Dosio and Fischer (2018) showed that precipitation will increase over most continental and northern parts of Europe (by +10-25% at 3 K), but to a lesser extent in summer (changes with 3 K between -5% at middle latitudes of Continental Europe to +10-15% at higher latitudes in the Boreal region), when natural and human demand are highest. As a result, short duration droughts could happen more frequently in some Eastern Europe catchments during summer even when supply does not change drastically due to the growth in natural demand (because of rising temperatures) and the contextual steady increase in human water demand for several socio-economic scenario (Ercin and Hoekstra, 2016). In the case of longer drought events, the imbalances between supply and demand over summer may

be mitigated by the increase in subsurface storages at the start of the summer season due to elevated
precipitation amounts during the previous seasons, but also potentially exacerbated in case of multi-
annual summer droughts. In this context, human induced factors may influence drought propagation
even further in high-regulated European basins (Van Loon et al., 2016).

**5.  Summary and Conclusions**

This study analysed how the main characteristics of hydrological droughts are expected to
change over Europe due to global warming. Projections in drought severity, duration and frequency
based on river water deficits highlight some common features and spatial patterns in future drought
conditions across Europe. The Mediterranean sub-region, which already suffers most from water
scarcity, is projected to experience the strongest effects of climate change on drought conditions.
With increasing global warming, streamflow deficits in this region are expected to happen more
frequently, become more severe and last longer. In contrast, the Boreal sub-area is projected to face
a consistent decrease in drought severity, duration and frequency.
In the Atlantic and Continental sub-regions the projections are less uniform, although over most
of the Atlantic drought conditions are projected to worsen, while they generally will become less
intense over Continental Europe. Despite the use of a large ensemble of climate models, there is still
a substantial uncertainty in the projections in these regions, even if changes at 3 K are mostly
statistically robust. The uncertainty is bigger for the 1.5 and 2 K GWLs, which suggests that there is
still large disagreement among the models in possible changes in drought conditions in these areas
when warming could be stabilised at the targets set in the Paris climate agreement. Since the climate
signal is less marked over these two sub-regions, projected water demand may play a more relevant
role in the direction of the future changes here. While in this study we considered water use
projections consistent with EU demographic, economic and energy projections, global and regional
water use studies show the large variability in future water use depending on the socioeconomic
scenario and water use model (Graham et al., 2018; Wada et al., 2016). Hence, apart from the
effects of warming on the hydrological cycle and natural water availability, socioeconomic
dynamics and consequent demand for water could also locally affect drought conditions.
Overall, the general patterns observed in this study are in line with the patterns observed in
studies that focused on specific temporal horizons rather than warming levels (Forzieri et al., 2014;
Spinoni et al., 2018; Stahl et al., 2012). Our study shows that with higher warming the changes in
drought traits are expected to be more marked, even if the spatial patterns of the areas with
increasing/decreasing drought conditions are rather similar for the three GWLs analysed here. The
outcomes obtained for different traits of streamflow droughts (i.e., severity, duration and frequency)
are in agreement with the results of Marx et al. (2018) based on the simple daily streamflow
percentile, suggesting again a strong coherence in streamflow climate projections.
The exposure analysis with population density and agricultural land highlights how at lower
warming levels positive and negative changes in exposure are expected to be balanced across
Europe. However, at higher GWLs the increase in population and agricultural land exposed in the
southern and western parts of Europe is projected to outweigh the effects of less severe droughts in
the less populated north and most of continental and eastern Europe. At 3 K warming this unbalance
between south-west and north-east could result in an additional 11 million people and 4.5 million ha
exposed each year to drought conditions that currently are expected to happen once every 10 years
or less frequently. The projected changes in exposure to drought will pose considerable challenges
for agriculture and water provision in densely populated and economically pivotal areas, especially
in southern Europe, making the findings of this study relevant to provide information that can be
used as a basis to evaluate the implications at European scale of climate mitigation policies.

**Data availability.** All data are freely available to the public via the EDO web portal
(https://edo.jrc.ec.europa.eu/) upon request. The main outputs of the study will be made available
through the JRC-DRMKC Risk Data Hub (https://drmkc.jrc.ec.europa.eu/risk-data-hub).

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

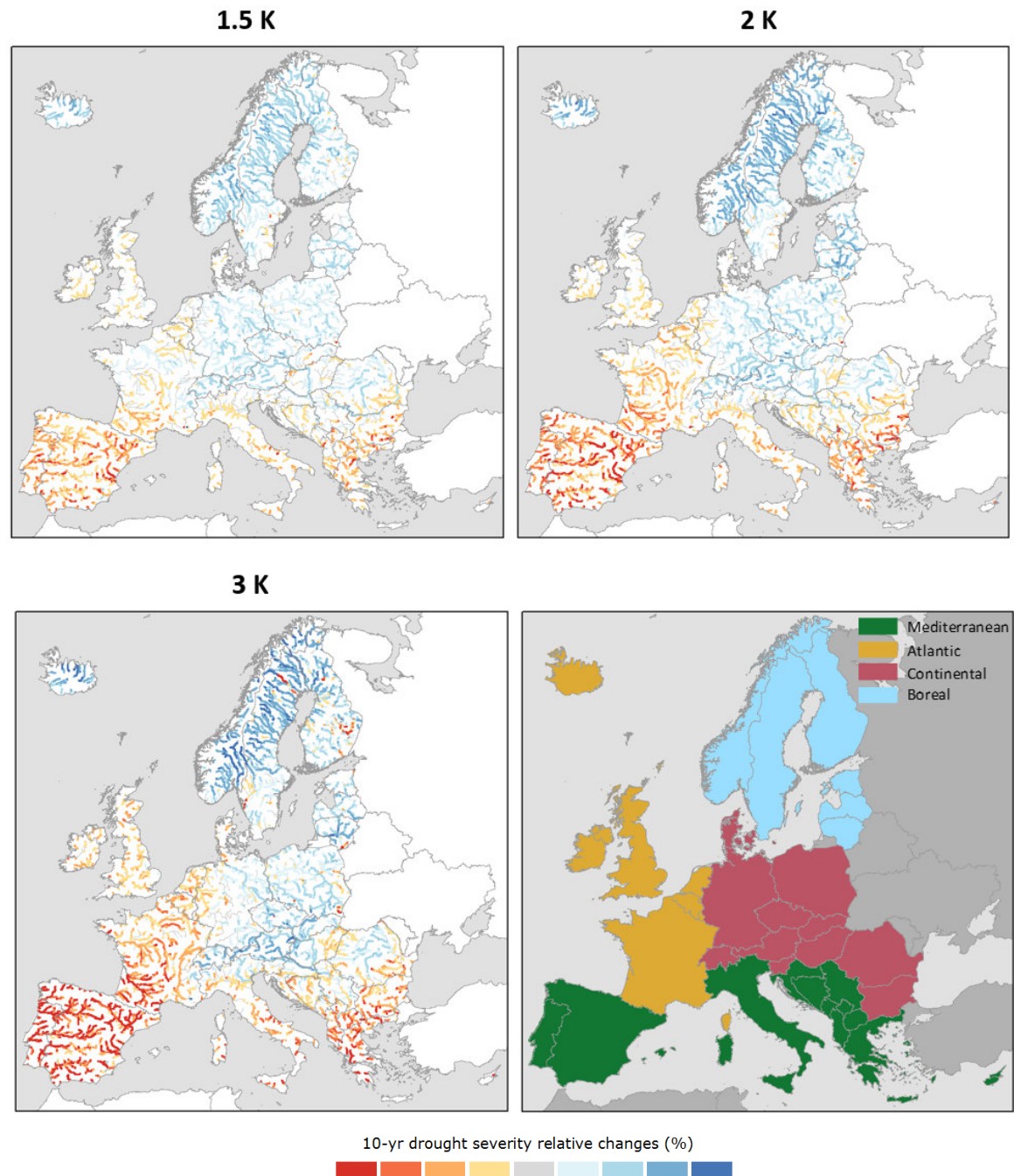

**Fig. 1.** Spatial distribution of the ensemble-median relative changes in drought severity of a 10-year drought (%) between reference period and the three GWLs (1.5 K in the upper-left panel, 2 K in the upper-right panel, 3 K in the lower-left panel). Positive values represent an increase in drought severity with warming. The no-agreement (no-agr) class identifies the cells where less than 2/3 of the climate ensemble members agree on the sign of the change. The lower-right panel represents the four sub-regions used for aggregation, which are in line with the IPCC AR5 European macro regions (Kovats et al., 2014).

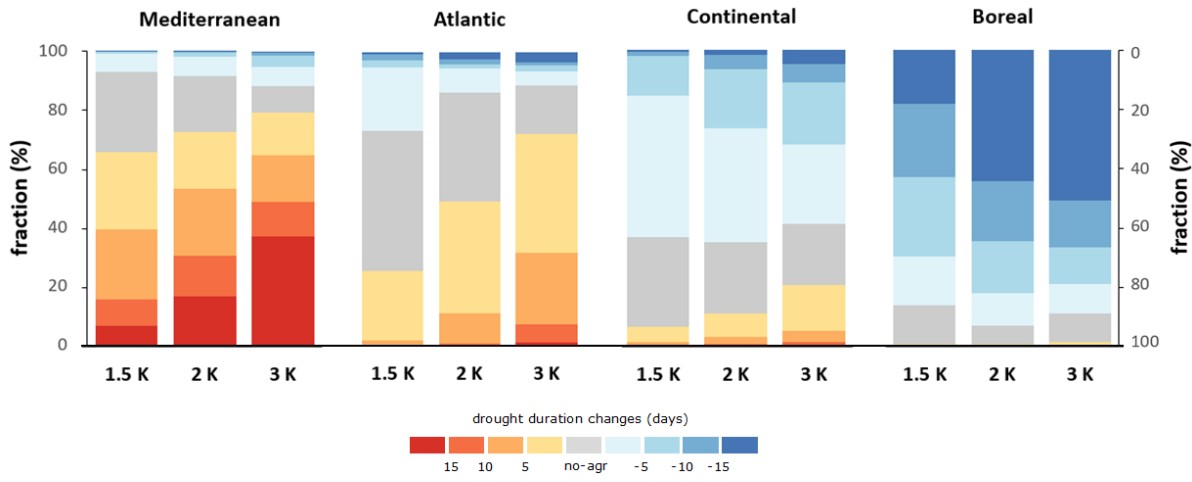


**Fig. 2.** Fraction of each sub-region within ranges of change in drought duration (days) for different
GWLs. Note that two y-axes are added to the figure only to facilitate the interpretation of the
positive (left axis) and negative (right axis) fraction values.

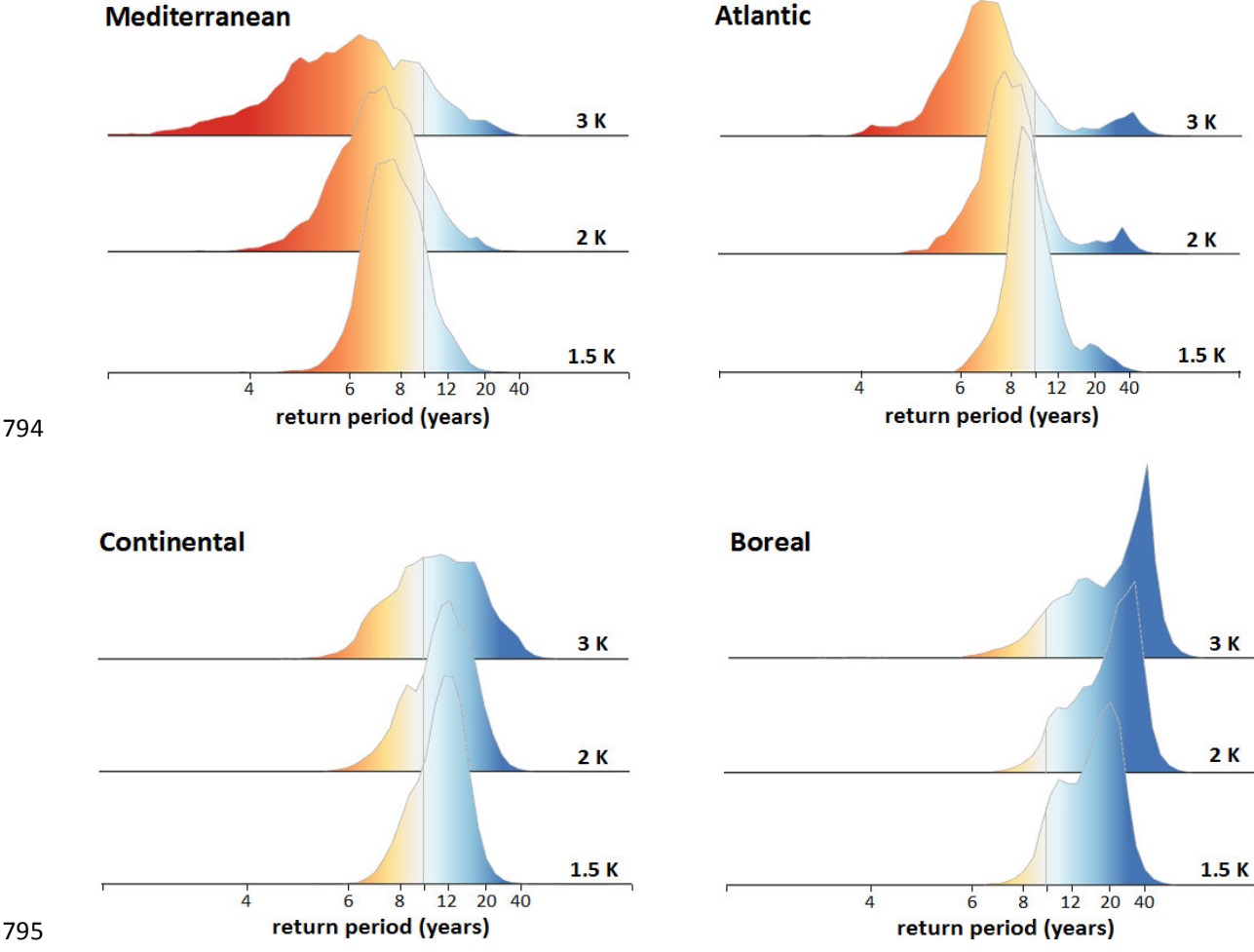



**Fig. 3.** Frequency distribution of the return period (years) for different GWLs corresponding to an event with a return period of 10 years in the reference baseline. Values lower (higher) than 10 represent an increase (reduction) in drought frequency. The vertical grey lines demark the 10-year return period, and the tick marks are uniformly spaced in frequency.

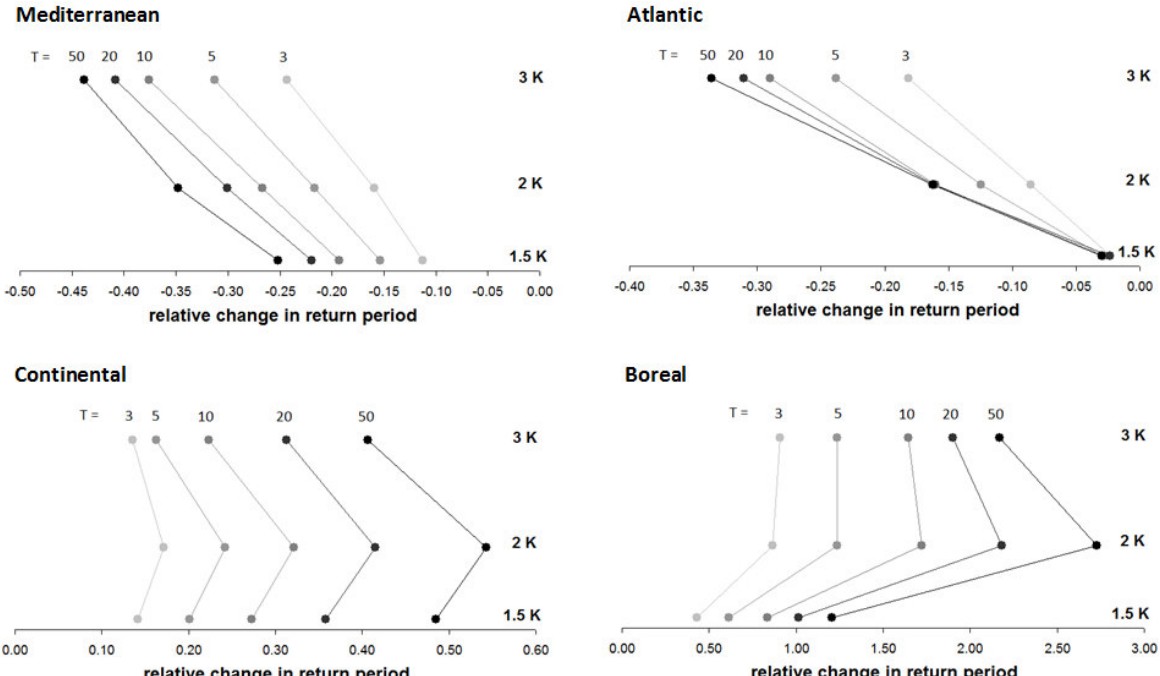


**Fig. 4.** Relative changes in sub-regional median return period (years) for different GWLs
corresponding to events with a return period of 3, 5, 10, 20 and 50 years in the reference baseline.
Negative (positive) values represent an increase (reduction) in drought frequency. Note that the x-
axis scale is different for each plot.

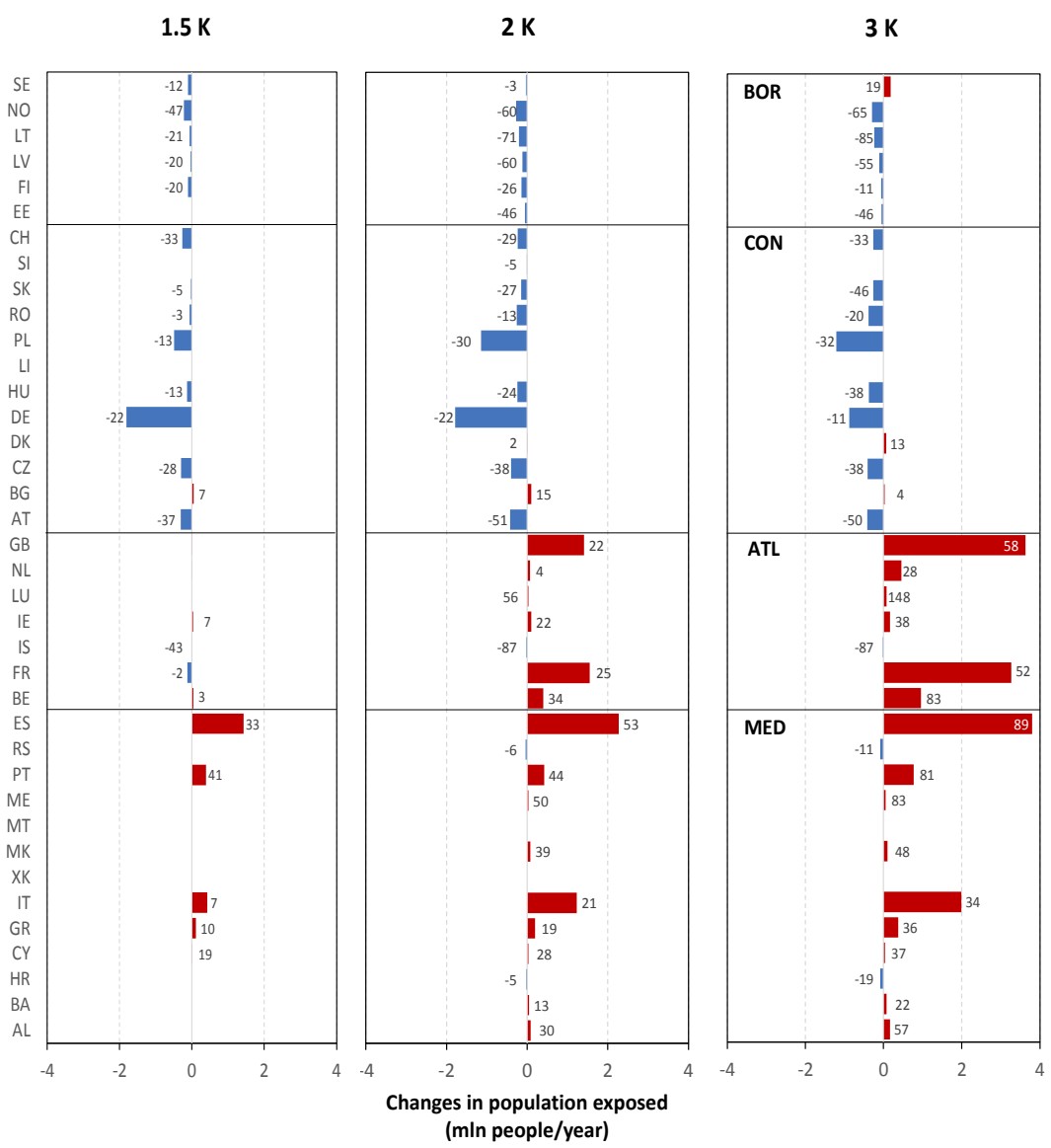

**Fig. 5.** Changes in population exposed per country (million people/year). Positive values indicate an increase in the population exposed. The numbers near the bars represent the percentage changes relative to the baseline (only if greater than 1%).

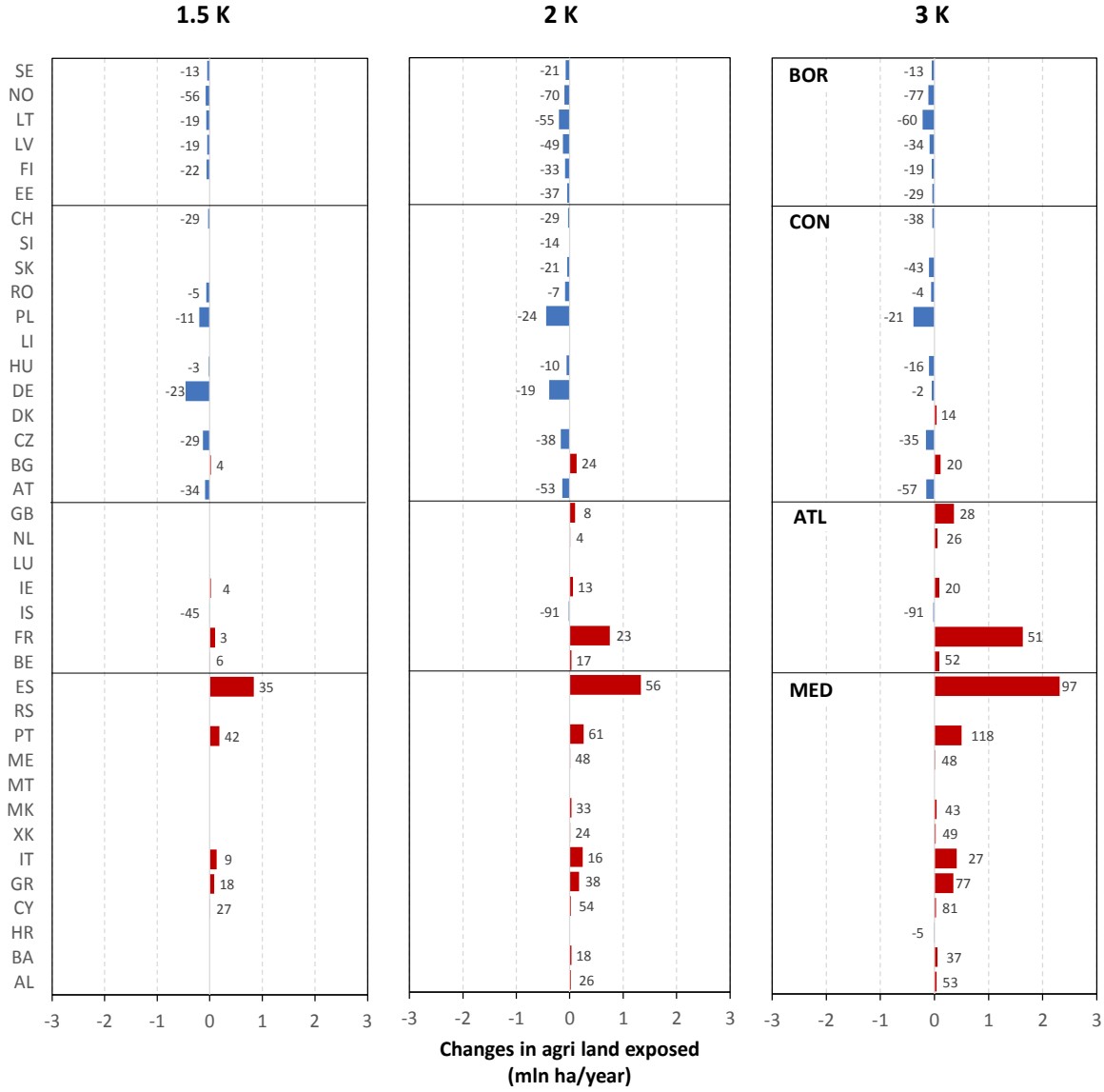


**Fig. 6.** Changes in agricultural land exposed per country (million ha/year). Positive values indicate
an increase in the area exposed. The numbers near the bars represent the percentage changes
relative to the baseline (only if greater than 1%).
**Table 1.** Total population exposed per sub-regions (million people/year).

| Name | baseline | 1.5 K | 2 K | 3 K |
|---|---|---|---|---|
| MEDITERRANEAN | 14.4 | 16.8 | 18.8 | 21.7 |
| ATLANTIC | 16.0 | 16.1 | 19.5 | 24.5 |
| CONTINENTAL | 19.6 | 16.2 | 15.0 | 15.5 |
| BOREAL | 2.5 | 2.0 | 1.7 | 1.9 |
| TOTAL | 52.5 | 51.1 | 55.0 | 63.6 |


**Table 2.** Total agricultural land exposed per sub-regions (million ha/year).

| Name | baseline | 1.5 K | 2 K | 3 K |
|---|---|---|---|---|
| MEDITERRANEAN | 5.8 | 7.1 | 8.0 | 9.6 |
| ATLANTIC | 5.4 | 5.5 | 6.3 | 7.6 |
| CONTINENTAL | 7.7 | 6.8 | 6.5 | 6.8 |
| BOREAL | 1.6 | 1.3 | 0.9 | 1.0 |
| TOTAL | 20.5 | 20.6 | 21.7 | 25.0 |
