# Peer review of "Diverging hydrological drought traits over Europe with global warming"

_Hydrology and Earth System Sciences, 2020_

## Referee Comment (RC1) · Anonymous Referee #1 · 4 Apr 2020

**Review for manuscript "Diverging hydrological drought traits over Europe with global warming"**

**Authors:** C. Cammalleri, G. Naumann, L. Mentaschi, B. Bisselink, E. Gelati, A. Roo, and L. Feyen
**Journal:** Hydrology and Earth System Sciences

**Summary**

Cammalleri et al. present a study on low flows/droughts in Europe under three different global warming levels (1.5, 2, and 3K temperature increase). They use the hydrological model LISFLOOD to simulate streamflow for a reference period (1981-2010) and for future 30-year periods corresponding to the three warming levels using an ensemble of future climate projections as model input. The model setup allows for the consideration of human water abstraction from the hydrological cycle. They determine the drought characteristics deficit, duration, and frequency for the reference and the future periods using a fixed drought threshold at the 15th flow percentile. Drought frequency is assessed using univariate return periods with respect to the deficit variable. The characteristics of the future periods are compared to the ones of the reference period. The study shows that the response of droughts to warming differs regionally within Europe. The Mediterranean and Atlantic regions are projected to be affected by longer, more severe, and more frequent droughts in future, while the Continental and Boreal regions are expected to experience shorter, less severe, and less frequent drought events. In addition to the drought characteristics assessment, the authors present an assessment of drought exposure with respect to population and agricultural area showing that the Mediterranean region needs to expect an increase in exposure both in agricultural land and population while the opposite is the case for the Continental region.

**General comments**

The study presented by Cammalleri et al. addresses a societally relevant question, i.e. how does global warming affect droughts in Europe with respect to duration, deficit, and frequency? While the study in itself is well motivated, the novelty of the approach could be made clearer and I see potential ways of extending the analyses into domains which have so far not received as much attention. I see many parallels to the study by [*Marx et al.*, 2018] who studied low flow characteristics under different global warming levels for Europe. The main advancement of this study compared to the study by [*Marx et al.*, 2018] are in my view threefold: (1) the authors use a drought definition instead of a simple low flow index which allows them to look at different drought characteristics including deficit and duration; (2) their model allows for the consideration of human flow modifications; and (3) they combine the hazard with an exposure analysis. I would make this clear in the introduction and clearly state what the added value of considering these three aspects is. In my opinion, the study presented could gain in profile, if the authors intensified the analysis of these aspects. Point 3 is probably easiest to tackle. They authors could highlight the exposure analysis in the introduction as this is something which goes beyond what previous studies have done. Points 1 and 2 could profit from some additional analyses. Regarding point 1, I would find a bivariate frequency analysis of deficit and duration interesting. Regarding point 2, it would be very interesting to show how drought characteristics change in a human-modified world as opposed to a world where such modifications are not considered (i.e. run model with and without the water use module and compare the changes in drought characteristics resulting from the different model runs). While the results of this study are well presented and tell a nice story, the methods section is in my opinion

very vague and it is hard to judge how suitable the model strategy is with respect to the analysis presented. The methods section would profit from specifications regarding model calibration and evaluation (was it calibrated at all?), an evaluation of the model simulations regarding the two drought characteristics deficit and duration (is the model able to well reproduce the phenomena studied?), a description of how the water demand estimates for the different sectors considered were derived (how was the disaggregation done?), and more information on the climate projections used. I think that this study will be a nice contribution to documenting future changes in drought characteristics once/if the validity of the methodology is clearly demonstrated and the novelty of the paper is clearly worked out.

**Specific comments**

Introduction: I would strengthen the two novel aspects of the study and use them as a motivation for the study: (1) the drought modeling considers water use and (2) the future evolution of drought exposure is assessed. I would also address the topic of drought definition and already point out here that you are using a fixed threshold to define droughts.

Methods: The methods section is in my opinion very vague and it is hard to judge the validity of the results in the absence of methodological detail. I suggest to address the following questions by making specifications accordingly:

1. Which quantile mapping approach was used? (l.102)
2. What is the reason for using the dataset EOBSv10 as an observed dataset? (l.103)
3. Was the assumption that 'between-pathway differences are generally much smaller than the within-pathway variability' verified for the drought characteristics analyzed? This assumption does not seem to be very intuitive to me. (107-109)
4. Was the LISFLOOD model calibrated, if so, how? If not, why not? (l.114)
5. How was the LISFLOOD model evaluated for the drought characteristics under study? Some evaluation plots are in my opinion required to prove the suitability of the model setup for the analysis performed (e.g. distribution of simulated vs. observed drought durations and deficits).
6. Is it correct that no downstream routing is performed but that the results presented are based on a grid-by-grid analysis of local streamflow generation? Please specify. (l.119-120)
7. How is irrigation water demand estimated? I.e. what does 'dynamically' mean and how was crop transpiration estimated? (122-123)
8. How was water demand estimated for the industrial, energy, livestock, and the domestic sectors? I.e. what was downscaled, how, and to which resolution (5*5km)? (125-126)
9. How are economic, budgetary, and demographic projections assumed to affect the individual water demand sectors considered? (l.126-128)
10. How does the Territorial Modelling Platform perform the downscaling? (129-131)
11. Why were the population and land use projections assumed to be static after 2050? (l.131-132).
12. What is the reason for the threshold choice? I assume that a fixed threshold is used? See e.g. [*Thiel et al.*, 2018] (l.138-139)
13. Did you do any smoothing to ensure the independence of events as e.g. suggested in [*Tallaksen and Hisdal*, 1997].

14. Why did you choose the Pareto Type II distribution to model drought deficits instead of the commonly used Generalized Pareto Distribution for partial duration series [*Coles*, 2001] and why is the threshold zero? Some goodness-of-fit test is required here (e.g. Anderson-Darling [*Chernobai et al.*, 2015]) (l.158-159).

15. How was the return period defined? (l. 163-165) The definition of a univariate return period of $T = \frac{1}{1-p}$ is valid when using annual maxima or annual minima time series. In the case of partial duration series as identified through a threshold level approach, the definition is $T = \frac{\mu}{1-p}$, where $\mu$ is the mean inter-arrival time between events (see e.g. [*Gräler et al.*, 2013; *Brunner et al.*, 2016]).

16. I am lost in the sentence in l.184-186. We would expect a 10-yearly event to occur on average every 10 years. This would expose all the people in the corresponding region to the event once every 10 years on average. How do you go to the assumption that one tenth of the population per NUTS 2 region is affected every year? I do not see the reasoning here because droughts are mostly larger scale phenomena and we can expect that most people in a region will be exposed at the same time instead of one $10^{th}$ of the population being exposed every year.

Results: The figures are clear and the results well presented. I think that the results section would profit from a display of the 'reference' situation and the seasonality of droughts over Europe (especially to highlight that drought seasonality using a fixed threshold will in Alpine regions happen during winter). The language used is pretty deterministic even though the results of projections are presented. I would rephrase sentences such as 'will increase', 'will last',… to something expressing that these results are uncertain e.g. 'are projected to increase', 'are expected to last',… Furthermore, it would just be interesting to present a few more results. Here, some suggestions for further analyses:

1. It would be interesting to see Figure 3 for two more return periods (e.g. 5 and 50 years) representing more frequent and rarer events, respectively to see how changes in frequency depend on the magnitude of events.
2. It would be interesting to look at drought duration return periods and at bivariate return periods of deficits and durations.
3. It would be very nice if the model could be run another time without the human water use component/module to illustrate the impact of human impact on future changes in drought characteristics. Adding this aspect would make this a truly novel analysis.

The study reads generally well but would still profit from editing.

**Minor points**

- Title: I personally would use the word 'characteristics' instead of 'traits'. This comment applies to the whole manuscript.
- l.14: I would not talk about an index in this case. I would already point out in the abstract that you are looking at drought characteristics derived using a threshold-level-approach with a fixed threshold.
- l.15: I would mention the model name already in the abstract.
- l.22: by 'interested', do you mean 'characterized'?

- l.23: specify reduction in what? Drought durations, deficits, and frequency.
- l.27: by 'this', do you refer to 'the regions most affected by changes'?
- Keywords: I would add LISFLOOD and global warming level.
- l.41: Yes, but the drought definition chosen also depends on the question at hand/problem of interest.
- l.41-43: the sentence seems incomplete. Suggested rephrasing: droughts are commonly looked at from a meteorological (), agricultural (), or hydrological () perspective.
- l. 44-47: I agree that there are more studies on meteorological drought than on soil moisture or streamflow drought. But there are many more potential examples for hydrological drought studies, e.g. [Hao and Aghakouchak, 2014; Laaha et al., 2017; Brunner et al., 2019].
- l.48: by 'This', do you refer to the smaller number of non-precipitation drought based studies?
- l.48: I would challenge the statement 'meteorological drought indicators have lower input data requirements than streamflow or soil moisture drought indicators'. If one would like to compute the Standardized runoff index [*Shukla and Wood*, 2008] instead of the SPI, a streamflow instead of a precipitation time series is needed, which is the same amount of data, i.e. one time series.
- l.49: specify 'this'. The focus on meteorological drought?
- l.52-54: cite the European Drought Impact Report Inventory (EDII) here?
- l.60-62: sentence would profit from rephrasing.
- l. 62: There are some studies that have looked at drought characteristics on a European scale and expected changes, e.g. [*Marx et al.*, 2018; *Samaniego et al.*, 2018; *Brunner and Tallaksen*, 2019].
- l.64-69: I would split this long sentence into two.
- l. 74: is there a reference documenting this paradigm shift?
- l. 82: 90th percentile of annual minima? or of annual mean? Do you actually mean the 10-th percentile with respect to non-exceedance probabilities? The 90th percentile is more commonly used for floods but I am aware that the drought and flood communities sometimes follow different conventions. Statistically, however, it would be more correct to talk about the 10th percentile.
- l.89: which mitigation targets are you referring to here?
- l. 90: water 'availability' instead of water 'budget'?
- l. 92: I would specify the model name.
- l.90-95: split long sentence into two.
- L. 93: I would add a reference to [*Moss et al.*, 2010].
- L.102: Euro-CORDEX initiative.
- L.168-171: I would move this information to the introduction.
- 173-174: this statement is not very true for hydropower production, which mostly happens in mountainous areas which are not very densely populated. And it is neither true for ecological purposes which can also be highly impacted by droughts but not considered in this study.
- L.177: could you shortly describe the properties of the LUISA projections?
- L.178: what is the average scale of these NUTS 2 areas?
- L.182: do you mean to refer to a '10-yearly' event, i.e. an event with a return period of 10 years?
- Figure 3: I would add a vertical line at 10 years as a reference, e.g. in light grey.
- L.205:209: I would move this information to the introduction of the methods section.
- L.223: I would indicate the Seine river basin on one of the maps (for non-European readers).

- L.354-355: start new sentence?
- L.356: reduction in drought severity and frequency?
- L. 391-394: could you clarify this sentence?

**References used in this review**

Brunner, M. I., and L. M. Tallaksen (2019), Proneness of European catchments to multiyear streamflow droughts, *Water Resour. Res.*, *55*, 8881– 8894, doi:10.1029/2019WR025903.

Brunner, M. I., J. Seibert, and A.-C. Favre (2016), Bivariate return periods and their importance for flood peak and volume estimation, *Wire's Water*, *3*, 819–833, doi:10.1002/wat2.1173.

Brunner, M. I., K. Liechti, and M. Zappa (2019), Extremeness of recent drought events in Switzerland: Dependence on variable and return period choice, *Nat. Hazards Earth Syst. Sci.*, *19*(10), 2311– 2323, doi:10.5194/nhess-19-2311-2019.

Chernobai, A., S. T. Rachev, and F. J. Fabozzi (2015), Composite goodness-of-fit tests for left-truncated loss samples, in *Handbook of financial econometrics and statistics*, edited by C.-F. Lee and J. Lee, pp. 575–596, Springer Science+Business Media, New York.

Coles, S. (2001), *An introduction to statistical modeling of extreme values*, Springer, London.

Gräler, B., M. J. van den Berg, S. Vandenberghe, A. Petroselli, S. Grimaldi, B. De Baets, and N. E. C. Verhoest (2013), Multivariate return periods in hydrology: a critical and practical review on synthetic design hydrograph estimation, *Hydrol. Earth Syst. Sci.*, *17*, 1281–1296, doi:10.5194/hess-17-1281-2013.

Hao, Z., and A. Aghakouchak (2014), A nonparametric multivariate multi-index drought monitoring framework, *J. Hydrometeorol.*, *15*(1), 89–101, doi:10.1175/JHM-D-12-0160.1.

Laaha, G. et al. (2017), The European 2015 drought from a hydrological perspective, *Hydrol. Earth Syst. Sci.*, *21*(6), 3001–3024, doi:10.5194/hess-21-3001-2017.

Marx, A., R. Kumar, S. Thober, O. Rakovec, N. Wanders, M. Zink, E. F. Wood, M. Pan, J. Sheffield, and L. Samaniego (2018), Climate change alters low flows in Europe under global warming of 1.5, 2, and 3 °C, *Hydrol. Earth Syst. Sci.*, *22*(2), 1017–1032, doi:10.5194/hess-22-1017-2018.

Moss, R. H. et al. (2010), The next generation of scenarios for climate change research and assessment, *Nature*, *463*(7282), 747–756, doi:10.1038/nature08823.

Samaniego, L., S. Thober, R. Kumar, N. Wanders, O. Rakovec, M. Pan, M. Zink, J. Sheffield, E. F. Wood, and A. Marx (2018), Anthropogenic warming exacerbates European soil moisture droughts, *Nat. Clim. Chang.*, *8*(5), 421–426, doi:10.1038/s41558-018-0138-5.

Shukla, S., and A. W. Wood (2008), Use of a standardized runoff index for characterizing hydrologic drought, *Geophys. Res. Lett.*, *35*(2), 1–7, doi:10.1029/2007GL032487.

Tallaksen, L. M., and H. Hisdal (1997), Regional analysis of extreme streamflow drought duration and deficit volume, *Friend'97 - Reg. Hydrol. Concepts Model. Sustain. Water Resour. Manag.*, *246*, 141–150, doi:10.1212/WNL.0b013e31823ed0a4.

Thiel, M. Van, A. J. Teuling, N. Wanders, M. J. P. Vis, K. Stahl, and A. F. Van Loon (2018), The role of glacier changes and threshold definition in the characterisation of future streamflow droughts in glacierised catchments, *Hydrol. Earth Syst. Sci.*, *22*(1), 463–485, doi:10.5194/hess-22-463-2018.

---

## Author Comment (AC1) · 30 Apr 2020

**Reply to anonymous Reviewer #1**

**General Comments**

We would like to tank the reviewer for his/her thoughtful revision of the manuscript. Here we provide some brief replies to his/her main comments, outlining the edits that we are planning to make in the revised version of the manuscript.

*The study presented by Cammalleri et al. addresses a societally relevant question, i.e. how does global warming affect droughts in Europe with respect to duration, deficit, and frequency? While the study in itself is well motivated, the novelty of the approach could be made clearer and I see potential ways of extending the analyses into domains which have so far not received as much attention. I see many parallels to the study by [Marx et al., 2018] who studied low flow characteristics under different global warming levels for Europe. The main advancement of this study compared to the study by [Marx et al., 2018] are in my view threefold: (1) the authors use a drought definition instead of a simple low flow index which allows them to look at different drought characteristics including deficit and duration; (2) their model allows for the consideration of human flow modifications; and (3) they combine the hazard with an exposure analysis. I would make this clear in the introduction and clearly state what the added value of considering these three aspects is.*

We thank the reviewer for properly identifying the main novelties of the study, and we plan to revisit the introduction to make them more evident.

*In my opinion, the study presented could gain in profile, if the authors intensified the analysis of these aspects. Point 3 is probably easiest to tackle. They authors could highlight the exposure analysis in the introduction as this is something which goes beyond what previous studies have done.*

We will further expand the focus on the exposure analysis in the introduction.

*Points 1 and 2 could profit from some additional analyses. Regarding point 1, I would find a bivariate frequency analysis of deficit and duration interesting.*

We are indeed exploring the possibility to focus future researches on a multi-variate analysis of different drought characteristics. Drought deficit and duration, however, are typically strongly correlated, hence a bivariate analysis of these two indicators would likely not deviate strongly from the analysis presented in our study. We think that a proper multi-variate analysis is worth of a full paper dedicated to the topic.

*Regarding point 2, it would be very interesting to show how drought characteristics change in a human-modified world as opposed to a world where such modifications are not considered (i.e. run model with and without the water use module and compare the changes in drought characteristics resulting from the different model runs).*

We agree with the reviewer that the effect of human water use is relevant for the analysis of drought. This is also why we considered this in our analysis. Forzieri et al. (2014)

showed in detail how water use alters river flows and streamflow drought indicators in different regions of Europe performing the analysis as suggested by the reviewer. In this study, a more detailed modeling of the dynamic socioeconomic condition is included, focusing on different aspects, namely on understanding drought hazard and exposure in a future world in case of climate inaction and different mitigation targets (warming levels). We believe that in order to address these questions a dynamic socioeconomic setting based on EU demographic, economic and budgetary projections is more appropriate and worth of the full focus of the paper.

*While the results of this study are well presented and tell a nice story, the methods section is in my opinion very vague and it is hard to judge how suitable the model strategy is with respect to the analysis presented. The methods section would profit from specifications regarding model calibration and evaluation (was it calibrated at all?), an evaluation of the model simulations regarding the two drought characteristics deficit and duration (is the model able to well reproduce the phenomena studied?), a description of how the water demand estimates for the different sectors considered were derived (how was the disaggregation done?), and more information on the climate projections used.*

The methods section was kept intentionally at minimum in order to maintain the focus of the readers on the results of the analysis. The LISFLOOD hydrological model that we used in this study has been applied in several pan-European studies on hydrology and climate. There is a large literature dedicated to several aspects of the modelling chain, including its calibration and validation. We will expand this section and better incorporate the relevant literature to accommodate your concerns.

*I think that this study will be a nice contribution to documenting future changes in drought characteristics once/if the validity of the methodology is clearly demonstrated and the novelty of the paper is clearly worked out.*

**Specific comments**

*Introduction: I would strengthen the two novel aspects of the study and use them as a motivation for the study: (1) the drought modeling considers water use and (2) the future evolution of drought exposure is assessed. I would also address the topic of drought definition and already point out here that you are using a fixed threshold to define droughts.*

We will reshape the introduction to better highlight the novelties of the study.

*Methods: The methods section is in my opinion very vague and it is hard to judge the validity of the results in the absence of methodological detail. I suggest to address the following questions by making specifications accordingly… (follow a list of 16 questions).*

We will incorporate the reviewer's suggestion by expanding the methods section, but also by better highlighting the relevant literature on each detail.

*Results: The figures are clear and the results well presented. I think that the results section would profit from a display of the 'reference' situation and the seasonality of droughts over Europe (especially to highlight that drought seasonality using a fixed threshold will in Alpine regions happen during winter).*

*The language used is pretty deterministic even though the results of projections are presented. I would rephrase sentences such as 'will increase', 'will last',... to something expressing that these results are uncertain e.g. 'are projected to increase', 'are expected to last',...*

We agree that projections of climate and consequently drought characteristics are uncertain. We will revisit the text to remove the instances where a deterministic language is misused.

*Furthermore, it would just be interesting to present a few more results. Here, some suggestions for further analyses:*

*1. It would be interesting to see Figure 3 for two more return periods (e.g. 5 and 50 years) representing more frequent and rarer events, respectively to see how changes in frequency depend on the magnitude of events.*

We observed that there is a rather strong relationship between the results at different return periods, as we where considering to incorporate the results for other return periods in the manuscript. We will add a figure summarizing these results, without replicating the same figure for different return periods, which may be too redundant and break the flow of the text.

*2. It would be interesting to look at drought duration return periods and at bivariate return periods of deficits and durations.*

As detailed above, we agree on the interest of the topic, but we consider the subject worth of a full paper that is currently under consideration.

*3. It would be very nice if the model could be run another time without the human water use component/module to illustrate the impact of human impact on future changes in drought characteristics. Adding this aspect would make this a truly novel analysis.*

As discussed above, this topic has been explored by other research studies, albeit with a less sophisticated modeling of socioeconomic conditions (e.g., Forzieri et al., 2014). Here we prefer to focus on the expected impact and exposure in case of climate inaction and different mitigation when the dynamic socioeconomic conditions are modeled at the best of our possibilities.

---

## Referee Comment (RC2) · Anonymous Referee #2 · 1 May 2020

This study examines the projected change in hydrologic drought severity, duration, and frequency due to climate change across Europe. It employs a unique GWL perspective to merge projections and represents a significant effort to combine climate, land cover, and population projections with hydrologic modeling to estimate drought exposure.

Overall the work is of a high quality; however, I have a number of reservations, as described below. The majority of these issues are clarifications of the methodology, which are needed to fully assess the findings. It is also important to clarify the interpretation of some results. I therefore recommend a significant revision.

Major issues:

M1. Is it possible to provide the range of years the ensemble members reach the GWLs

for context? It would help to confirm that the present conditions have not surpassed 1.5K and provide some context to how fare off +1.5K is from the present. If this is not possible, at least provide delta K for the reference period.

M2. Related to A1, you are incorporating changes in population, land cover, and water abstraction with time through 2050. But, because the endpoints are tied to GWL, rather than a year, each member of your ensemble will have slightly different values for these model inputs. Are you accounting for this? Can you provide a relative estimate of the water abstraction changes? This would help provide sensitivity/scale for this portion of the model.

M3. Changes in snowmelt patterns and seasonality have a potential impact on future hydrologic changes at higher elevations and latitudes. You mention this on Line 372. Does your model incorporate a snow accumulation/melt module?

M4. Please provide more clarification as to how the return periods are being derived. More detail is needed than the reference to Cammalleri et al (2017) paper. It appears you are using a peak-over-threshold/partial duration series approach. I am most familiar with using the generalized pareto distribution for return periods in this context. It appears like you are using the Pareto Type II. Please explain this choice. Also, be aware that in the context of a partial duration series, your statement on line 163 "the probability that one event is topped in any one year" is slightly less accurate than for an annual maximum series.

M5. Please provide the methodology for calculating the change in drought duration shown in Figure 2. Does days/year represent a summation of all drought days during the reference period? I believe this is the correct interpretation. My confusion is because the Severity (D) analysis focuses on the severity of an individual event, whereas this Duration analysis focuses on a cumulative metric.

Also, as part of this, please revise your interpretation in Section 3.1.2. If you are summing up the days under drought conditions, then you cannot say that "droughts

will last longer", as you do in Line 252. I interpret longer droughts as the individual drought events lasting longer, but this metric could increase due to more frequent, but similar duration droughts. Without knowing the number of unique droughts, you cannot make this statement, only that the total time spent in drought will increase.

M6. There is no significance testing for any of these claims. It is difficult to determine whether these trends are a significant signal or noise. The consistent regional patterns suggest a true trend. But, I would strongly recommend significance testing to quantify how much agreement there is among ensemble models (Fig 1) or how significant these changes are regionally (Fig 2/3).

M7. Line 426 - This interpretation, which depends on your assumption on Line 184, assumes independence among sites, which is not true. Regions enter drought at the same time, so it is not fair to say that 10% of the region will be exposed to a 10 year drought in any given year. More likely, a majority of the Mediterranean (or at least the eastern/western portions) will enter drought at the same time.

Associated with this is the interpretation of Figure 4/5. Is this based on the 10-year drought only or all droughts?

M8. Please provide a data availability statement. This is required by HESS and is not included in the version I had access to.

Minor issues:

- You are defining your GWLs relative to a pre-industrial baseline. Please provide the years for this baseline. Is it the 1881–1910 baseline used in Donnelly et al. (2017)?

- Line 160 - If you are using Maximum Likelihood to fit the Lomax distribution, this is not an "empirical" cumulative distribution, but rather an estimate of the population's cumulative distribution.

- Figure 1 - This figure caption and legend do not indicate that this is showing the change in the 10-year drought.
- Line 196 - Please indicate where these macro regions were derived from.

- Line 241 - I suggest you use "climate change-induced" here. Much of this trend is likely driven by changes in precipitation, rather than warming specifically. Similarly, on Line 423.

- Figure 3 - What is this x-axis? Is it standard normal deviates spacing? There isn't quite enough tick marks to know for sure. Can you please explain this in the caption?

- Figure 3 - Please add some type of reference point to this figure to highlight the 10 year drought event, as defined by the reference period. In its current format, there is not even a label fo the 10 year event. At a minimum, add this label, preferably add a vertical line so the reader can compare with the plotted distributions.

- Line 345 - You may also consider adding the following references, which provide additional support for this regional pattern of meteorological drought. They both attempt to parse the affect of precipitation trends from temperature/evapotranspiration trends.

Dubrovská, M., Hayes, M., Duce, P., Trnka, M., Svoboda, M., & Zara, P. (2013). Multi-GCM projections of future drought and climate variability indicators for the Mediterranean region. Regional Environmental Change, 14(5), 1907–1919. doi:10.1007/s10113-013-0562-z

Stagge, J.H., Kingston, D.G., Tallaksen, L.M. et al. Observed drought indices show increasing divergence across Europe. Sci Rep 7, 14045 (2017). https://doi.org/10.1038/s41598-017-14283-2

Line 349 - The word "severe" is misspelled.

---

## Author Response (AR1)

**Reply to anonymous Reviewer #1**

**General Comments**

We would like to thank the reviewer for his/her thoughtful revision of the manuscript. We hope that we were able to address the major concerns of the reviewer in the revised version of the manuscript.

*The study presented by Cammalleri et al. addresses a societally relevant question, i.e. how does global warming affect droughts in Europe with respect to duration, deficit, and frequency? While the study in itself is well motivated, the novelty of the approach could be made clearer and I see potential ways of extending the analyses into domains which have so far not received as much attention. I see many parallels to the study by [Marx et al., 2018] who studied low flow characteristics under different global warming levels for Europe. The main advancement of this study compared to the study by [Marx et al., 2018] are in my view threefold: (1) the authors use a drought definition instead of a simple low flow index which allows them to look at different drought characteristics including deficit and duration; (2) their model allows for the consideration of human flow modifications; and (3) they combine the hazard with an exposure analysis. I would make this clear in the introduction and clearly state what the added value of considering these three aspects is.*

We revisited the introduction to make more evident these three key points.

*In my opinion, the study presented could gain in profile, if the authors intensified the analysis of these aspects. Point 3 is probably easiest to tackle. They authors could highlight the exposure analysis in the introduction as this is something which goes beyond what previous studies have done.*

We have expanded the focus on the exposure analysis in the introduction.

*Points 1 and 2 could profit from some additional analyses. Regarding point 1, I would find a bivariate frequency analysis of deficit and duration interesting.*

We agree that exploring a multi-variate analysis of different drought characteristics is an interesting future researches topic. Drought deficit and duration, however, are typically strongly correlated, hence a bivariate analysis of these two indicators would likely not deviate strongly from the analysis presented in our study. We think that a proper multi-variate analysis is worth of a full paper dedicated to the topic.

*Regarding point 2, it would be very interesting to show how drought characteristics change in a human-modified world as opposed to a world where such modifications are not considered (i.e. run model with and without the water use module and compare the changes in drought characteristics resulting from the different model runs).*

We agree with the reviewer that the effect of human water use is relevant for the analysis of drought. This is also why we considered this in our analysis. Forzieri et al. (2014) showed in detail how water use alters river flows and streamflow drought indicators in different regions of Europe performing the analysis as suggested by the reviewer. In this study, a more detailed modeling of the dynamic socioeconomic conditions is included, focusing on different aspects, namely on understanding drought hazard and exposure in a future world in case of climate inaction and different mitigation targets (warming levels). We believe that in order to address these questions a dynamic socioeconomic setting based on EU demographic, economic and budgetary projections is more appropriate and worth of the full focus of the paper.

*While the results of this study are well presented and tell a nice story, the methods section is in my opinion very vague and it is hard to judge how suitable the model strategy is with respect to the analysis presented. The methods section would profit from specifications regarding model calibration and evaluation (was it calibrated at all?), an evaluation of the model simulations regarding the two drought characteristics deficit and duration (is the model able to well reproduce the phenomena studied?), a description of how the water demand estimates for the different sectors considered were derived (how was the disaggregation done?), and more information on the climate projections used.*

We expanded the methodology section to address reviewer's main questions (see specific comments). However, since the LISFLOOD hydrological model has been extensively used/tested in several pan-European studies on hydrology, climate and drought we referred to the relevant literature where needed in order to keep the section concise.

*I think that this study will be a nice contribution to documenting future changes in drought characteristics once/if the validity of the methodology is clearly demonstrated and the novelty of the paper is clearly worked out.*

We hope that the new version of the manuscript better highlighted the novelties of the study.

**Specific comments**

*Introduction: I would strengthen the two novel aspects of the study and use them as a motivation for the study: (1) the drought modeling considers water use and (2) the future evolution of drought exposure is assessed. I would also address the topic of drought definition and already point out here that you are using a fixed threshold to define droughts.*

We modified the introduction to better highlight the novelties of the study.

*Methods: The methods section is in my opinion very vague and it is hard to judge the validity of the results in the absence of methodological detail. I suggest to address the following questions by making specifications accordingly:*

*1. Which quantile mapping approach was used? (L.102)*

*2. What is the reason for using the dataset EOBSv10 as an observed dataset? (L.103)*

The forcing dataset was produced by Dosio (2020) in the framework of the PESETA 4 project (https://ec.europa.eu/jrc/sites/jrcsh/files/pesetaiv_task_1_climate_final_report.pdf), as it was not specifically made only for this study. Detailing the bias correction is out of the scope of this paper, but we clarified the relevant reference and source in the new version of the manuscript.

*3. Was the assumption that 'between-pathway differences are generally much smaller than the within-pathway variability' verified for the drought characteristics analyzed? This assumption does not seem to be very intuitive to me. (L.107-109)*

This is a result of the recent study published by Mentaschi et al. (2020) on the same dataset and on the annual minimum (drought), average and maximum flow (flood). Tests were also made for severity but not included in the publication. In this study it is shown the independence of changes at a certain warming level from the adopted pathway. We reworded the sentence to clarify.

*4. Was the LISFLOOD model calibrated, if so, how? If not, why not? (L.114)*

Yes, the model has been calibrated as part of its operational implementation in EFAS (https://www.efas.eu/). We added the corresponding reference to the new version of the manuscript.

*5. How was the LISFLOOD model evaluated for the drought characteristics under study? Some evaluation plots are in my opinion required to prove the suitability of the model setup for the analysis performed (e.g. distribution of simulated vs. observed drought durations and deficits).*

The model has been evaluated specifically for drought at both European and Global scale as part of its implementation for operational drought monitoring in the European and Global Drought Observatories (https://edo.jrc.ec.europa.eu/edov2/php/index.php?id=1000). We better clarified this in the section 2.3 of the new version of the manuscript.

*6. Is it correct that no downstream routing is performed but that the results presented are based on a grid-by-grid analysis of local streamflow generation? Please specify. (L.119-120).*

The model has a routing component, which is described in the methodology: "the surface runoff generated in each cell is channeled to the nearest river network cell by means of a routing component based on a 4-point implicit finite-difference solution of the kinematic wave (Chow et al., 1988)". We slightly reworded the sentence to avoid any miscommunication.

*7. How is irrigation water demand estimated? I.e. what does 'dynamically' mean and how was crop transpiration estimated? (L.122-123)*

We added some detailed on the supply-demand approach used for the irrigation modeling in crops and constant water level for paddy-rice. As for all the other sectors, further details can be found in Bisselink et al. (2018).

*8. How was water demand estimated for the industrial, energy, livestock, and the domestic sectors? I.e. what was downscaled, how, and to which resolution (5\*5km)? (L.125-126)*

Data at country level were obtained from sources like EUROSTAT, and then downscaled to the LISFLOOD grid using different techniques and proxy variables. More details on the downscaling are reported in the new version of the manuscript.

*9. How are economic, budgetary, and demographic projections assumed to affect the individual water demand sectors considered? (L.126-128)*

According to the downscaling procedure used for each sector, the future projections were used as proxy for the downscaling of the future water uses. We completely re-organized this section to better clarify the procedure.

*10. How does the Territorial Modelling Platform perform the downscaling? (L.129-131)*

The high resolution data from the LUISA platform were used to downscale water demand for the different sectors. We re-elaborated the entire paragraph to clarify.

*11. Why were the population and land use projections assumed to be static after 2050? (L.131-132).*

Data for the LUISA platform are available until 2050. We clarified this in the new version of the manuscript.

*12. What is the reason for the threshold choice? I assume that a fixed threshold is used? See e.g. [Thiel et al., 2018] (L.138-139).*

The use of the historical threshold for the future projections is a widely adopted approach, aiming at evaluating how present day droughts will be perceived in the future. In this framework, a transient threshold will not be suitable for such analysis. We clarified this assumption in the new version of the manuscript as "derived for the present climate as a threshold both in the present and future scenarios, with the aim to estimate how present condition droughts will be projected under climate change".

*13. Did you do any smoothing to ensure the independence of events as e.g. suggested in [Tallaksen and Hisdal, 1997].*

We did not perform smoothing on the data, but we applied pooling of consecutive close events (Zelenhasić and Salvai, 1987) and removal of isolated minor events (Jakubowski and Radczuk, 2004) to ensure both the independence of events and the absence of distortion in the fitting through minor events. We clarified the role of these two procedures in the revised version of the manuscript.

*14. Why did you choose the Pareto Type II distribution to model drought deficits instead of the commonly used Generalized Pareto Distribution for partial duration series [Coles, 2001] and why is the threshold zero? Some goodness-of-fit test is required here (e.g. Anderson-Darling [Chernobai et al., 2015]) (L.158-159).*

The Lomax distribution is just a special case of the Generalized Pareto Distribution (GPD), when the μ parameter is set to 0. We found this distribution to perform adequately at global scale (see Cammalleri et al. 2020, where a proper goodness-of-fit test is performed) for drought deficit, since this variable is limited at zero as lower threshold. We reworded the section to clarify how the distribution has been previously tested in another study.

*15. How was the return period defined? (L. 163-165) The definition of a univariate return period of T=1/1−p is valid when using annual maxima or annual minima time series. In the case of partial duration series as identified through a threshold level approach, the definition is T=μ/1−p', where μ is the mean inter-arrival time between events (see e.g. [Gräler et al., 2013; Brunner et al., 2016]).*

This is correct. We applied the correct definition of the return period, as now clarified in the revised definition of the return period in the text (including a reference to Serinaldi, 2015).

*16. I am lost in the sentence in L.184-186. We would expect a 10-yearly event to occur on average every 10 years. This would expose all the people in the corresponding region to the event once every 10 years on average. How do you go to the assumption that one tenth of the population per NUTS 2 region is affected every year? I do not see the reasoning here because droughts are mostly larger scale phenomena and we can expect that most people in a region will be exposed at the same time instead of one 10th of the population being exposed every year.*

We notice that this section caused misunderstanding for both reviewers, and we revisited the text to clarify the goal of this part of the study. Your interpretation is correct, and we agree that droughts usually occur over large areas, hence it is likely that all population will be affected at the same time rather than 1/10 every year.

Here, we estimate the expected average annual exposure in the 30-year periods, which is a theoretical expected exposure that would occur in any given year if exposure from all drought probabilities and magnitudes are spread out equally over time (here those with return period of 10 years or less frequent). As correctly pointed out by the reviewer, this does not mean that each year has the same exposure to drought. Rather, in some years there will be high exposure, while in (most) others there will be low or no exposure.

*Results: The figures are clear and the results well presented. I think that the results section would profit from a display of the 'reference' situation and the seasonality of droughts over Europe (especially to highlight that drought seasonality using a fixed threshold will in Alpine regions happen during winter).*

*The language used is pretty deterministic even though the results of projections are presented. I would rephrase sentences such as 'will increase', 'will last',… to something expressing that these results are uncertain e.g. 'are projected to increase', 'are expected to last',…*

We agree that projections of climate and consequently drought characteristics are uncertain. We were careful in trying to convey this uncertainty in our discussion, but we revisited the text to remove the instances where a deterministic language is misused.

*Furthermore, it would just be interesting to present a few more results. Here, some suggestions for further analyses:*

*1. It would be interesting to see Figure 3 for two more return periods (e.g. 5 and 50 years) representing more frequent and rarer events, respectively to see how changes in frequency depend on the magnitude of events.*

We observed that there is a rather strong relationship between the results at different return periods. We added a figure summarizing these results (Figure 4 in the new version of the manuscript), without replicating the same figure for different return periods, which may be too redundant and break the flow of the text.

*2. It would be interesting to look at drought duration return periods and at bivariate return periods of deficits and durations.*

As detailed above, we agree on the interest of the topic, but we consider the subject worth of a full paper that is currently under consideration.

*3. It would be very nice if the model could be run another time without the human water use component/module to illustrate the impact of human impact on future changes in drought characteristics. Adding this aspect would make this a truly novel analysis.*

As discussed above, this topic has been explored by other research studies, albeit with a less sophisticated modeling of socioeconomic conditions (e.g., Forzieri et al., 2014). Here we focus on the expected impact and exposure in case of climate inaction and different mitigation when the dynamic socioeconomic conditions are modeled at the best of our possibilities.

*The study reads generally well but would still profit from editing.*

During the revision we have carefully checked the paper throughout.

**Minor points**

- *Title: I personally would use the word 'characteristics' instead of 'traits'. This comment applies to the whole manuscript.*

We have used traits in other related studies, so we prefer to leave the title as it is, since this is not a major correction.

- *L.14: I would not talk about an index in this case. I would already point out in the abstract that you are looking at drought characteristics derived using a threshold-level-approach with a fixed threshold.*

We agree and reworded the abstract accordingly.

- *L.15: I would mention the model name already in the abstract.*

Done.

- *L.22: by 'interested', do you mean 'characterized'?*

Done.

- *L.23: specify reduction in what? Drought durations, deficits, and frequency.*

Clarified.

- *L.27: by 'this', do you refer to 'the regions most affected by changes'?*

Yes, we reworded the sentence to clarify.

- *Keywords: I would add LISFLOOD and global warming level.*

Done.

- *L.41: Yes, but the drought definition chosen also depends on the question at hand/problem of interest.*

- *L.41-43: the sentence seems incomplete. Suggested rephrasing: droughts are commonly looked at from a meteorological (), agricultural (), or hydrological () perspective*

We reworded the paragraph as: "Depending on the degree of penetration of the water deficit into the hydrological cycle, drought is commonly classified into meteorological (e.g., precipitation), agricultural (e.g., soil moisture) and hydrological (e.g., river discharge) drought (Wilhite, 2000). Each class of drought may be seen more relevant depending on the specific application, and different effects of climate change are likely to be observed depending on the corresponding analysed indicators (Feng, 2017)".

- *L.44-47: I agree that there are more studies on meteorological drought than on soil moisture or streamflow drought. But there are many more potential examples for hydrological drought studies, e.g. [Hao and Aghakouchak, 2014; Laaha et al., 2017; Brunner et al., 2019].*

We agree that there are many more examples in the literature (and the same is true for meteorological drought). Here we reported only few examples of studies on climate projection of drought at continental scale. We rephrased to clarify that.

We also added the reference to Brunner et al. (2019) in the discussion on regional/local studies.

- *L.48: by 'This', do you refer to the smaller number of non-precipitation drought based studies?*

Reworded.

- *L.48: I would challenge the statement 'meteorological drought indicators have lower input data requirements than streamflow or soil moisture drought indicators'. If one would like to compute the Standardized runoff index [Shukla and Wood, 2008] instead of the SPI, a streamflow instead of a precipitation time series is needed, which is the same amount of data, i.e. one time series.*

We reworded the sentence to clarify our point.

- *L.49: specify 'this'. The focus on meteorological drought?*

Done.

- *L.52-54: cite the European Drought Impact Report Inventory (EDII) here?*

Done.

- *L.60-62: sentence would profit from rephrasing.*

Done.

- *L.62: There are some studies that have looked at drought characteristics on a European scale and expected changes, e.g. [Marx et al., 2018; Samaniego et al., 2018; Brunner and Tallaksen, 2019].*

We referred to some studies, including Marx et al. (2018), in the next paragraph. Keep in mind that here we are discussing only hydrological drought at this point, hence the missing reference to Samaniego et al. (2018) (cited early in te text).

- *L.64-69: I would split this long sentence into two.*

Done.

- *L.74: is there a reference documenting this paradigm shift?*

This shift is a consequence of the focus in the Paris agreement, , where a target to limit global warming to well below 2 degrees Celsius. This make more relevant to analyze warming levels rather than specific emission target at given years (i.e., Kyoto protocol). We do not think that there is a more relevant reference than the Paris agreement itself, already cited in the text.

- *L.82: 90th percentile of annual minima? or of annual mean? Do you actually mean the 10-th percentile with respect to non-exceedance probabilities? The 90th percentile is more commonly used for floods but I am aware that the drought and flood communities sometimes follow different conventions. Statistically, however, it would be more correct to talk about the 10th percentile.*

They used the 90th percentile of exceedance, as now clarified in the text. We prefer to keep this definition to be consistent with the original paper.

- *L.89: which mitigation targets are you referring to here?*

We reworded to clarify the connection to the Paris agreement.

- *L.90: water 'availability' instead of water 'budget'?*

Done.

- *L.92: I would specify the model name.*

Done.

- *L.90-95: split long sentence into two.*

The sentence was reworded.

- *L.93: I would add a reference to [Moss et al., 2010].*

Done.

- *L.102: Euro-CORDEX initiative.*

Done.

- *L.168-171: I would move this information to the introduction.*

We agree that this paragraph was out of place. We partially move this information in the expanded section of the introduction dedicated to the exposure analysis, and reworded this paragraph to harmonize the content.

- *L.173-174: this statement is not very true for hydropower production, which mostly happens in mountainous areas which are not very densely populated. And it is neither true for ecological purposes which can also be highly impacted by droughts but not considered in this study.*

We reworded to clarify the reasoning behind our approach. Also, we clarified in the introduction how ecological impacts are not considered in this study.

- *L.177: could you shortly describe the properties of the LUISA projections?*

Some details were added, including an additional reference to the full description of the platform. A description of the platform is out of the scope of the paper.

- *L.178: what is the average scale of these NUTS 2 areas?*

NUTS2 regions vary country by country (e.g., in Germany correspond to the Regierungsbezirke, in Italy the Regioni and in UK the Counties). By definition, on average, they have between 800,000 and 3,000,000 inhabitants.

- *L.182: do you mean to refer to a '10-yearly' event, i.e. an event with a return period of 10 years?*

Yes, we reworded to clarify.

- *Figure 3: I would add a vertical line at 10 years as a reference, e.g. in light grey.*

We added the vertical lines to demark the 10-year return period.

- *L.205:209: I would move this information to the introduction of the methods section.*

This information was derived from the results of the analysis and we prefer to keep it here to avoid confusion in the flow of the text (i.e., reference to this figure in the methods section). However, we reshaped the paragraph in order to clarify how the macro-regions were derived, following the suggestion of the other reviewer.

- *L.223: I would indicate the Seine river basin on one of the maps (for non-European readers).*

We reworded the text to clarify the spatial location of the river basin.

- *L.354-355: start new sentence?*

We reworded the sentence.

- *L.356: reduction in drought severity and frequency?*

Fixed.

- *L.391-394: could you clarify this sentence?*

We reshaped the full paragraph to improve the clarity of the message.

**References used in this review**

*Brunner, M. I., and L. M. Tallaksen (2019), Proneness of European catchments to multiyear streamflow droughts, Water Resour. Res., 55, 8881– 8894, doi:10.1029/2019WR025903.*

Brunner, M. I., J. Seibert, and A.-C. Favre (2016), Bivariate return periods and their importance for flood peak and volume estimation, Wire's Water, 3, 819–833, doi:10.1002/wat2.1173.

Brunner, M. I., K. Liechti, and M. Zappa (2019), Extremeness of recent drought events in Switzerland: Dependence on variable and return period choice, Nat. Hazards Earth Syst. Sci., 19(10), 2311–2323, doi:10.5194/nhess-19-2311-2019.

Chernobai, A., S. T. Rachev, and F. J. Fabozzi (2015), Composite goodness-of-fit tests for left-truncated loss samples, in Handbook of financial econometrics and statistics, edited by C.-F. Lee and J. Lee, pp. 575–596, Springer Science+Business Media, New York.

Coles, S. (2001), An introduction to statistical modeling of extreme values, Springer, London.

Gräler, B., M. J. van den Berg, S. Vandenberghe, A. Petroselli, S. Grimaldi, B. De Baets, and N. E. C. Verhoest (2013), Multivariate return periods in hydrology: a critical and practical review on synthetic design hydrograph estimation, Hydrol. Earth Syst. Sci., 17, 1281–1296, doi:10.5194/hess-17-1281-2013.

Hao, Z., and A. Aghakouchak (2014), A nonparametric multivariate multi-index drought monitoring framework, J. Hydrometeorol., 15(1), 89–101, doi:10.1175/JHM-D-12-0160.1.

Laaha, G. et al. (2017), The European 2015 drought from a hydrological perspective, Hydrol. Earth Syst. Sci., 21(6), 3001–3024, doi:10.5194/hess-21-3001-2017.

Marx, A., R. Kumar, S. Thober, O. Rakovec, N. Wanders, M. Zink, E. F. Wood, M. Pan, J. Sheffield, and L. Samaniego (2018), Climate change alters low flows in Europe under global warming of 1.5, 2, and 3 °C, Hydrol. Earth Syst. Sci., 22(2), 1017–1032, doi:10.5194/hess-22-1017-2018.

Moss, R. H. et al. (2010), The next generation of scenarios for climate change research and assessment, Nature, 463(7282), 747–756, doi:10.1038/nature08823.

Samaniego, L., S. Thober, R. Kumar, N. Wanders, O. Rakovec, M. Pan, M. Zink, J. Sheffield, E. F. Wood, and A. Marx (2018), Anthropogenic warming exacerbates European soil moisture droughts, Nat. Clim. Chang., 8(5), 421–426, doi:10.1038/s41558-018-0138-5.

Shukla, S., and A. W. Wood (2008), Use of a standardized runoff index for characterizing hydrologic drought, Geophys. Res. Lett., 35(2), 1–7, doi:10.1029/2007GL032487.

Tallaksen, L. M., and H. Hisdal (1997), Regional analysis of extreme streamflow drought duration and deficit volume, Friend'97 - Reg. Hydrol. Concepts Model. Sustain. Water Resour. Manag., 246, 141–150, doi:10.1212/WNL.0b013e31823ed0a4.

*Thiel, M. Van, A. J. Teuling, N. Wanders, M. J. P. Vis, K. Stahl, and A. F. Van Loon (2018), The role of glacier changes and threshold definition in the characterisation of future streamflow droughts in glacierised catchments, Hydrol. Earth Syst. Sci., 22(1), 463–485, doi:10.5194/hess-22-463-2018.*

**Reply to anonymous Reviewer #2**

**General Comments**

*This study examines the projected change in hydrologic drought severity, duration, and frequency due to climate change across Europe. It employs a unique GWL perspective to merge projections and represents a significant effort to combine climate, land cover, and population projections with hydrologic modeling to estimate drought exposure. Overall the work is of a high quality; however, I have a number of reservations, as described below. The majority of these issues are clarifications of the methodology, which are needed to fully assess the findings. It is also important to clarify the interpretation of some results. I therefore recommend a significant revision.*

We thank the reviewer for the constructive comments on the manuscript. We hope that the major issues are now addressed in the revised version of the manuscript.

**Major issues**

*M1. Is it possible to provide the range of years the ensemble members reach the GWLs for context? It would help to confirm that the present conditions have not surpassed 1.5K and provide some context to how fare off +1.5K is from the present. If this is not possible, at least provide delta K for the reference period.*

We agree that this information will give more context to our results. We will provide the temperature difference between the preindustrial period and the baseline (delta K = +0.7K) and an indication of the ensemble variability in the years to reach GWLs for the two RCPs.

*M2. Related to A1, you are incorporating changes in population, land cover, and water abstraction with time through 2050. But, because the endpoints are tied to GWL, rather than a year, each member of your ensemble will have slightly different values for these model inputs. Are you accounting for this? Can you provide a relative estimate of the water abstraction changes? This would help provide sensitivity/scale for this portion of the model.*

We are indeed accounting for this, and the reviewer is correct that ensemble members may have slightly different values for some of the underlying socioeconomic variables. The projected changes in socioeconomic variables are available in 5-year time steps. Demographic and land use changes in Europe are relatively mild up to 2050, while they remain constant afterwards. Hence, spread in water abstraction driven by the socioeconomic drivers and the effect on water availability are small compared to the effects of climate change, with the latter also affecting water demand for crops irrigation.

We revisited the description of the water use modules, which now provides more details on the modelling procedure.

*M3. Changes in snowmelt patterns and seasonality have a potential impact on future hydrologic changes at higher elevations and latitudes. You mention this on Line 372. Does your model incorporate a snow accumulation/melt module?*

Yes, LISFLOOD has a snow module that is based on the degree-day factor method. We will better emphasize this in the model description.

*M4. Please provide more clarification as to how the return periods are being derived. More detail is needed than the reference to Cammalleri et al (2017) paper. It appears you are using a peak-over-threshold/partial duration series approach. I am most familiar with using the generalized pareto distribution for return periods in this context. It appears like you are using the Pareto Type II. Please explain this choice. Also, be aware that in the context of a partial duration series, your statement on line 163 "the probability that one event is topped in any one year" is slightly less accurate than for an annual maximum series.*

The Pareto Type II is a special case of the Generalized Pareto distribution, hence analogous considerations can be made (we now clarified this in the text). We agree with the reviewer that the statement on the return period can be confusing in our specific case for readers that are only familiar with annual min/max series. We revisited the text to clarify the definition and added a reference to a relevant publication.

*M5. Please provide the methodology for calculating the change in drought duration shown in Figure 2. Does days/year represent a summation of all drought days during the reference period? I believe this is the correct interpretation. My confusion is because the Severity (D) analysis focuses on the severity of an individual event, whereas this Duration analysis focuses on a cumulative metric.*

*Also, as part of this, please revise your interpretation in Section 3.1.2. If you are summing up the days under drought conditions, then you cannot say that "droughts will last longer", as you do in Line 252. I interpret longer droughts as the individual drought events lasting longer, but this metric could increase due to more frequent, but similar duration droughts. Without knowing the number of unique droughts, you cannot make this statement, only that the total time spent in drought will increase.*

The reviewer's interpretation of the definition of duration in the original version of the manuscript is correct. Following your considerations, we updated the figure by focusing on the duration of the event, and revisited the text accordingly. We agree that this quantity, rather than the total number of drought days in a year, is fitting better the rest of the analyses performed in the study.

*M6. There is no significance testing for any of these claims. It is difficult to determine whether these trends are a significant signal or noise. The consistent regional patterns suggest a true trend. But, I would strongly recommend significance testing to quantify*

*how much agreement there is among ensemble models (Fig 1) or how significant these changes are regionally (Fig 2/3).*

The robustness of the changes has been accounted by reporting the areas where at least 2/3 of the ensemble models agree on the sign of the change. The area with no-agreement (usually in grey) are the ones where this condition is not met. We better clarified this choice in the revised version of the text.

*M7. Line 426 - This interpretation, which depends on your assumption on Line 184, assumes independence among sites, which is not true. Regions enter drought at the same time, so it is not fair to say that 10% of the region will be exposed to a 10 year drought in any given year. More likely, a majority of the Mediterranean (or at least the eastern/western portions) will enter drought at the same time.*

*Associated with this is the interpretation of Figure 4/5. Is this based on the 10-year drought only or all droughts?*

We agree that drought usually occur over large areas, hence it is likely that all population will be affected at the same time rather than 1/10 every year. We estimate and present the expected average annual exposure for each 30-year period, which is the exposure that would occur in any given year if exposure from all drought probabilities and magnitudes are spread out equally over time (here those with return period of 10 years or less frequent). As correctly pointed out by the reviewer, this does not mean that each year has the same exposure to drought. Rather, in some years there will be high exposure, while in (most) others there will be low or no exposure. We understand that this caused some confusion, since it has been pointed out by both reviewers. We revisited the text to clarify this, and added a figure on the relationship between different return periods, as suggested by the other reviewer.

*M8. Please provide a data availability statement. This is required by HESS and is not included in the version I had access to.*

All the data produced by the JRC are freely available to the public upon request. We are also planning to disseminate some of the key outputs throughout our Risk Data Hub (https://drmkc.jrc.ec.europa.eu/risk-data-hub). We will add this information to the manuscript.

*Minor issues:*

*- You are defining your GWLs relative to a pre-industrial baseline. Please provide the years for this baseline. Is it the 1881–1910 baseline used in Donnelly et al. (2017)?*

Yes, we added this information to the new version of the manuscript.

*- Line 160 - If you are using Maximum Likelihood to fit the Lomax distribution, this is not an "empirical" cumulative distribution, but rather an estimate of the population's cumulative distribution.*

We were referring to the frequency distribution of D values before the fitting. We reworded to avoid any misunderstanding.

*- Figure 1 - This figure caption and legend do not indicate that this is showing the change in the 10-year drought.*

Thanks for point out this oversight. We modified both the caption and the legend to clarify that.

*- Line 196 - Please indicate where these macro regions were derived from.*

We reorganized this section, also following the suggestion of reviewer #1. Now we clarified how the regions were derived from, and how they were compared with the ones used by IPCC.

*- Line 241 - I suggest you use "climate change-induced" here. Much of this trend is likely driven by changes in precipitation, rather than warming specifically. Similarly, on Line 423.*

We agree on the change here. In Line 423 we replace with GWL since we are referring to the analyzed global warming level.

*- Figure 3 - What is this x-axis? Is it standard normal deviates spacing? There isn't quite enough tick marks to know for sure. Can you please explain this in the caption?*

The data are equally spaced in frequency, we added this information in the caption.

*- Figure 3 - Please add some type of reference point to this figure to highlight the 10 year drought event, as defined by the reference period. In its current format, there is not even a label of the 10 year event. At a minimum, add this label, preferably add a vertical line so the reader can compare with the plotted distributions.*

We added a vertical line to identify the 10 year frequency.

*- Line 345 - You may also consider adding the following references, which provide additional support for this regional pattern of meteorological drought. They both attempt to parse the affect of precipitation trends from temperature/evapotranspiration trends.*

*Dubrovský, M., Hayes, M., Duce, P., Trnka, M., Svoboda, M., & Zara, P. (2013). Multi-GCM projections of future drought and climate variability indicators for the Mediterranean region. Regional Environmental Change, 14(5), 1907–1919. doi:10.1007/s10113-013-0562-z*

*Stagge, J.H., Kingston, D.G., Tallaksen, L.M. et al. Observed drought indices show increasing divergence across Europe. Sci Rep 7, 14045 (2017). https://doi.org/10.1038/s41598-017-14283-2*

Thanks for suggesting those references. They were added to the new version of the manuscript.

*Line 349 - The word "severe" is misspelled.*

Fixed.

[revised manuscript text omitted]

---

## Referee Report (RR1)

**Review for manuscript "Diverging hydrological drought traits over Europe with global warming"**

**Authors:** C. Cammalleri, G. Naumann, L. Mentaschi, B. Bisselink, E. Gelati, A. Roo, and L. Feyen
**Journal:** Hydrology and Earth System Sciences

**Summary**

Cammalleri et al. present a study on low flows/droughts in Europe under three different global warming levels (1.5, 2, and 3K temperature increase). They use the hydrological model LISFLOOD to simulate streamflow for a reference period (1981-2010) and for future 30-year periods corresponding to the three warming levels using an ensemble of future climate projections as model input. The model setup allows for the consideration of human water abstraction from the hydrological cycle. They determine the drought characteristics deficit, duration, and frequency for the reference and the future periods using a fixed drought threshold at the 15$^{th}$ flow percentile. Drought frequency is assessed using univariate return periods with respect to the deficit variable. The characteristics of the future periods are compared to the ones of the reference period. The study shows that the response of droughts to warming differs regionally within Europe. The Mediterranean and Atlantic regions are projected to be affected by longer, more severe, and more frequent droughts in future, while the Continental and Boreal regions are expected to experience shorter, less severe, and less frequent drought events. In addition to the drought characteristics assessment, the authors present an assessment of drought exposure with respect to population and agricultural area showing that the Mediterranean region needs to expect an increase in exposure both in agricultural land and population while the opposite is the case for the Continental region.

**General comments**

I appreciate the additional methodological specification added to the methods section, which helped to clarify a few details. However, I still have one major concern regarding a proof for model suitability for drought analysis in Europe. I really think that presenting one summary figure of model performance in terms of different drought characteristics in the methods section would strengthen trust in the key messages of the paper. Such an evaluation seems particularly important because the model was calibrated and validated at a global scale with a focus on floods instead of droughts (l. 144-146). In addition, I have a few minor suggestions. I see further room for more explicitly working out the novelty of the paper in the introduction. Because the human demand projections seem to be the major contribution of this study, I would dedicate some more attention to them in the discussion part. I also suggest a few minor modifications to the Figures, which may facilitate reading them, and to put some additional effort into editing the paper with respect to sentence structure, the use of commas, and wording.

**Specific comments**

Introduction: I would still more explicitly state the two main aims of the study in the introduction: (1) quantify the impact of climate change on drought characteristics under three different global warming levels and (2) assess the effect of projected climate change on the population and agricultural land exposed to drought.

Methods: The methods section has considerably improved in clarity. However I would still expect some actual proof for the suitability of the model for drought analyses in Europe. As I suggested in my earlier review, this could be achieved by comparing observed to simulated drought characteristics (duration and deficit) for a set of example catchments. In addition, the description of the Lomax function needs to be revised.

Results: The figures are clear and the results well presented. I suggest some minor adjustments to the Figures.

Discussion: I think that the demand projections deserve some more attention in the discussion section because they distinguish this study from previous studies on future droughts in Europe. I would e.g. look at the contributions by [*Wada et al.*, 2016; *Graham et al.*, 2018] who look at future demand projections under different socio-economic pathways scenarios.

The study reads generally well on a paragraph level but would still profit from editing on a sentence level and from a consistent use of tense. I am going to make a few examples under 'suggested edits', however, this list is not exhaustive.

**Minor points**

- When talking about the analysis performed in this study I would consistently use the term drought instead of low flow (e.g. l. 14, l.182, l.184, l.189).
- I would appreciate a consistent use of tense when describing methods. L. 16 e.g. uses the past tense ('was') while l. 14 uses the present ('employ'). There are other instances in the text where tense is used inconsistently and I would pay particular attention to this aspect when editing the manuscript. Other examples are 'focuses' on l.120 and 'used' on l.116.
- A few phrases would profit from the specification of 'this' or 'these', which are sometimes used in isolation without a clear reference (e.g. l.54 'This is also highlights'). I suggest going through the manuscript and replacing these instances by more specific terms. On l.54 e.g. This focus on meteorological drought? Another example is: 'this issue' on l.92: which issue?
- L. 127-132: I would clearly highlight that this statement is an assumption because it is not intuitive and also not in line with some of the literature out there. I agree that this assumption is in some cases useful, I just think it should be openly declared as an assumption and not as a fact.
- L. 157: what about the non-member states such as Norway or Switzerland? They are still part of the analysis.
- L. 206-208: I think that this description of the Lomax function is not entirely correct. The Lomax function has 0 support and a mean of kappa/(alpha-1) and is only defined for alpha > 1. The statement that the 'location parameter is equal to zero' is therefore wrong. Please also provide a reference to the publication, where this distribution was first introduced.
- Would move l. 217-218 to l.211. Was this goodness-of-fit testing done for the same catchments used in this study (or a subset of them)?
- L. 289: It seems that there are just two river basins in Denmark which were actually considered in this analysis. Are they representative of the Danish hydrology? I would maybe refrain from specially mentioning it.
- Were livestock and domestic use kept constant in future? If so, why? If not, please shortly describe the estimation procedure used.

- L. 246: I agree with reviewer 2 that some significance testing would be highly desirable. If this is not done, please at least mention that no significance testing was done and that the definition of 'robustness' entirely relies on the sign but not the significance of change (l. 243-246).
- L. 442: I think this decrease in summer streamflow is not only due to less precipitation but also smaller snowmelt contributions see e.g. [*Stahl et al.*, 2016; *Jenicek et al.*, 2018].
- L. 452-454: I think that these statements need references. And as mentioned above, I think that this section about water use should be expanded.
- L. 477-482: I think that this information is partially redundant and could be merged with the introduction and methods.
- Data availability: please specify from whom the data can be requested and what subset will be made available through the JRC data hub.

**Figures**

- I would add lables (a), (b),… to all subfigures presented. This would facilitate referencing in the text.
- Figure 2: When I first looked at the figure, I was confused by the two inversed scales. I understand now why they are useful. Still, a note in the figure caption would be helpful.
- Figure 3: I would indicate the medians for all PDFs to facilitate following section 3.1.3. The tick marks are clearly not 'uniformly' spaced as indicated in the figure caption.
- Figure 4: I would indicate that the x-axis labels differ between subplots.
- Figure 5: I would indicate the country abbreviations on Figure 1d and I would limit the scale to +/- 4 in order to improve legibility and reduce white space.
- Figure 6: Similarly, I would limit these figures to +/- 3.
- Tables 1 and 2: I would write 'total' instead of 'tot'

**Suggested edits**

- L.23: suggest rephrasing to 'is expected to experience' instead of 'sees' to be less deterministic.
- I suggest replacing the keyword 'low-flow index' by 'human water use' and 'frequency analysis'.
- L.44-46: suggest rephrasing the sentence to something like: 'A specific drought type may be perceived most relevant for a given application and various indicators may experience different effects of climate change.'
- L. 49: suggest replacing 'climate projection of' to 'impact of climate change on'.
- L. 53: suggest replacing 'with the latter usually requiring' by 'whose analysis usually requires'.
- L. 66: maybe rather use 'domain' instead of 'extent'?
- L. 76: remove 'of' in front of 'past'.
- L. 81: has it already shifted or is it still shifting?
- L. 88: the word 'annual' confused me here. Was the threshold not determined using daily streamflow values?
- L. 90: remove 'a' in front of 'key'.
- L. 92: By 'this issue', do you mean 'future drought changes under the influence of climate change and water abstraction'?

- L. 95: would rephrase to: ' the threshold level method for event extraction, which allows for a detailed frequency analysis of different streamflow drought characteristics including severity, duration, and frequency.'
- L. 100: incorporate's'
- Would remove l.105-108 or merge with l.56-58 to avoid redundancy.
- L. 112-113: What is the purpose of this sentence in the introduction? Would more this to the methods section.
- L. 124: suggest replacing 'over' by 'compared to'.
- L. 126: what does 'on average' refer to? All model runs? Also suggest replacing 'at' by 'in'.
- L. 134: suggest adding 'a' in front of '5'.
- L. 140: snow accumulation and melt.
- L. 161: I wonder how relevant 'paddy-rice irrigation' is in Europe? If this is irrelevant in Europe, I would exclude its description (l. 164-165).
- L. 166: is 'a' function of…
- L. 172: what about non-urban but still populated areas?
- L. 173: corresponding to what?
- L.185: please mention that Q85 refers to exceedance probabilities.
- L. 190: 'volume' instead of 'area'?
- L. 192: suggest removing 'temporal'.
- L. 194-198: I would reorganize this sentence and swap 2) with 1) because of the order of the two elements in the first part of the sentence.
- L. 198-201: This information seems to be redundant with information provided on l.194-195 and can in my opinion be removed.
- L. 202: Following this 'drought' definition,..
- L. 203: 'was' derived. Would also remove 'huge'.
- L. 215-217: what is the purpose of this sentence here?
- L. 221-2023: I think this sentence needs rephrasing.
- L. 231-234: rephrasing recommended.
- L. 237: aggregate'd'.
- L. 251-255: I think this sentence needs rephrasing.
- L. 260: Atlantic 'region'.
- L. 266: replace 'halve' by 'half'.
- L. 287: the rest 'of the region' shows…
- L. 297: longer droughts 'with increasing GWL'.
- L. 300: longer 'than 5 days'. Applies to the whole section: longer than 'what'?
- L. 307: last longer 'than in the reference period'.
- L. 318-319: This statement does not seem to be true for the Boreal region at 3K.
- L. 324: Figure three seems to show densities not distributions.
- L. 357: clear'ly'
- L. 361: maybe specify, i.e. the rarer events.
- L. 403: 'change' patterns?
- L. 405-408: think this sentence needs rephrasing.
- L. 409: would replace 'temporal horizons' with 'time periods instead of GWL'.

- L. 414 and other instances: I would not use the word 'negative' and 'positive' because they don't seem to be used in an objective sense in terms of – and + but rather in the sense of perception. See also L. 431 'positive', which could be replaced by decreasing.
- L. 422: would replace 'symmetrically' by 'in contrast'.
- L. 425: 'are' instead of 'is'.
- L. 434: supply '(precipitation)'?
- L. 443: would delete 'for the Paris warming targets'.
- L. 447: What does 'this' refer to?
- L. 450: would delete 'months'.
- L. 467: region 'are' expected
- L. 468: would replace 'symmetrically' by 'in contrast'.
- L. 473: by 'mostly well defined', do you mean 'robust'?
- L. 487: would swap the order of 'here' and 'analysed'.
- L. 490: agricultural 'land' exposed in 'the' southern…
- L. 492: what does 'this' refer to?
- L. 494: less 'frequently'.

**References used in this review**

Graham, N. T. et al. (2018), Water sector assumptions for the Shared Socioeconomic Pathways in an integrated modeling framework, *Water Resour. Res.*, *54*(9), 6423–6440, doi:10.1029/2018WR023452.

Jenicek, M., J. Seibert, and M. Staudinger (2018), Modeling of future changes in seasonal snowpack and impacts on summer low flows in Alpine catchments, *Water Resour. Res.*, *54*(1), 538–556, doi:10.1002/2017WR021648.

Stahl, K., M. Weiler, I. Kohn, D. Freudiger, J. Seibert, M. Vis, and K. Gerlinger (2016), *The snow and glacier melt components of streamflow of the river Rhine and its tributaries considering the influence of climate change*, Freiburg.

Wada, Y. et al. (2016), Modeling global water use for the 21st century: The Water Futures and Solutions (WFaS) initiative and its approaches, *Geosci. Model Dev.*, *9*(1), 175–222, doi:10.5194/gmd-9-175-2016.

---

## Referee Report (RR2)

**Review for manuscript "Diverging hydrological drought traits over Europe with global warming"**

**Authors:** C. Cammalleri, G. Naumann, L. Mentaschi, B. Bisselink, E. Gelati, A. Roo, and L. Feyen
**Journal:** Hydrology and Earth System Sciences

**General comments**

While the authors adequately addressed most of the minor points risen in my previous review, my major point, i.e. the one on model validation, has in my opinion not been sufficiently addressed. The authors now cite two studies, which previously used a similar model setup as used in this study. I had a look at these two studies, which indeed demonstrate that that previous LISFLOOD model setup simulated drought deficits well. However, the present study seems to use a setup obtained using a different objective function than in these previous studies, which do not mention the Kling-Gupta efficiency (KGE) used here. I would be surprised if calibrating on the classical KGE resulted in good low flow performance because that metric puts more weight on high than low flows (see e.g. [Garcia et al., 2017]). Given the choice of this objective function, it seems even more crucial to show the reader that the simulated droughts have the same statistical characteristics as the observed droughts. I therefore still think that presenting one summary figure of model performance in terms of different drought characteristics in the methods section is required to strengthen trust in the key messages of the paper. Why is providing such a figure such a big deal?

**Specific comments**

Methods: The methods section has considerably improved in clarity. However I would still expect some actual proof for the suitability of the model for drought analyses in Europe. As I suggested in my earlier two reviews, this could be achieved by comparing observed to simulated drought characteristics (duration and deficit) for a set of example catchments.

The study reads generally well on a paragraph level but would still profit from editing on a sentence level and from a consistent use of tense. I am going to make a few examples under 'suggested edits', however, this list is not exhaustive.

**Minor points**

I would add lables (a), (b),… to all subfigures presented. This would facilitate referencing in the text. Panel labels are actually required according to the HESS manuscript submission guidelines, which say: 'Labels of panels must be included with brackets around letters being lower case (e.g. (a), (b), etc.).'

**Suggested edits:**

l. 14: 'drought' instead of 'low-flow'?
l. 84: 'provide' instead of 'provided'
l. 93-96: sentence would read much better in active mode.
l. 100: remove 'of'
l. 103: specify 'compartments' already up here.
l. 142: 'consists' instead of 'consist'.
l. 148: 'Specifically' instead of 'in detail'

l. 157: provide a reference to the 'literature'

l. 177: 'extend instead of 'go up to'

l. 215-223: section needs rephrasing because the term 'D events' suggest that the lomax function is fitted to the number of events, which is not the case as it was fitted to deficits (D). Suggestion: 'the series of deficits were fitted…'

l. 487: 'scenarios' instead of 'scenario'

**Reference used in this review**

Garcia, F., N. Folton, and L. Oudin (2017), Which objective function to calibrate rainfall–runoff models for low-flow index simulations?, *Hydrol. Sci. J.*, *62*(7), 1149–1166, doi:10.1080/02626667.2017.1308511.

---

## Author Response (AR2)

**Replies to Reviewer #1**

**General comments**

I appreciate the additional methodological specification added to the methods section, which helped to clarify a few details. However, I still have one major concern regarding a proof for model suitability for drought analysis in Europe. I really think that presenting one summary figure of model performance in terms of different drought characteristics in the methods section would strengthen trust in the key messages of the paper. Such an evaluation seems particularly important because the model was calibrated and validated at a global scale with a focus on floods instead of droughts (l. 144-146). In addition, I have a few minor suggestions. I see further room for more explicitly working out the novelty of the paper in the introduction. Because the human demand projections seem to be the major contribution of this study, I would dedicate some more attention to them in the discussion part. I also suggest a few minor modifications to the Figures, which may facilitate reading them, and to put some additional effort into editing the paper with respect to sentence structure, the use of commas, and wording.

We thank the reviewer for his/her thoughtful comments. We revised and expanded the section dedicated to the model calibration/validation by adding the following:
- a reference to the most recent calibration/validation exercise over Europe. While the name of the model may suggest that the model is suitable only for modelling floods, this is no longer the case. The model has been improved over the years in terms of conceptualization, data input and calibration, with the aim to model a range of hydrological variables beyond extreme high river flows. The current calibration procedure is based on the optimization of the Kling-Gupta Efficiency, which aims at fitting not only the peaks but the entire flow duration curve. The previous reference on the global calibration is kept as a reference to a detailed description of the calibration algorithm.
- References to previous studies where the same modeling framework is used for drought analyses, including a detailed validation against ground data in terms of both minimum flow and deficit.

In addition, in the section on drought modeling it is now better clarified how the index has been operationally implemented within the EDO monitoring system, validated against past relevant drought events, as well as currently used in operation drought reports on major droughts over Europe.

We hope that this evidence is sufficient to confirm the overall reliability of the modeling framework in the context of droughts.

Point to point responses to the other comments are provided in the next sections.

**Specific comments**

Introduction: I would still more explicitly state the two main aims of the study in the introduction: (1) quantify the impact of climate change on drought characteristics under three different global warming levels and (2) assess the effect of projected climate change on the population and agricultural land exposed to drought.

We revised the last section of the introduction to further stress the two main goals.

Methods: The methods section has considerably improved in clarity. However I would still expect some actual proof for the suitability of the model for drought analyses in Europe. As I suggested in my earlier review, this could be achieved by comparing observed to simulated drought characteristics (duration and deficit) for a set of example catchments. In addition, the description of the Lomax function needs to be revised.

We expanded the section on the model calibration/validation, and we added further details on the use of the modeling framework in the context of drought monitoring. Reference on recent validation of the Lisflood specifically for minimum flow and deficit were also highlighted.

Results: The figures are clear and the results well presented. I suggest some minor adjustments to the Figures.

Please see the section "minor points" for details on the adjustments made to the figures.

Discussion: I think that the demand projections deserve some more attention in the discussion section because they distinguish this study from previous studies on future droughts in Europe. I would e.g. look at the contributions by [*Wada et al.*, 2016; *Graham et al.*, 2018] who look at future demand projections under different socio-economic pathways scenarios.

We agree that we overlooked the potentially large variability in future water use depending on socioeconomic scenario and water use model. We now better acknowledge this in the discussion, including the references suggested by the reviewer.

The study reads generally well on a paragraph level but would still profit from editing on a sentence level and from a consistent use of tense. I am going to make a few examples under 'suggested edits', however, this list is not exhaustive.

We thoroughly revised the manuscript to improve the editing and consistency.

**Minor points**

• When talking about the analysis performed in this study I would consistently use the term drought instead of low flow (e.g. l. 14, l.182, l.184, l.189).

Across the manuscript the term low-flow is only used in these few instances, and it always has a really specific meaning (i.e. low-flow index in EDO, which is the name used for the indicator, low-flow analysis, which cannot be simply replaced by drought analysis). We ensure that in the rest of the manuscript the term drought is always used instead.

• I would appreciate a consistent use of tense when describing methods. L. 16 e.g. uses the past tense ('was') while l. 14 uses the present ('employ'). There are other instances in the text where tense is used inconsistently and I would pay particular attention to this aspect when editing the manuscript. Other examples are 'focuses' on l.120 and 'used' on l.116.

We thoroughly revised the method section to remove inconsistencies.

• A few phrases would profit from the specification of 'this' or 'these', which are sometimes used in isolation without a clear reference (e.g. l.54 'This is also highlights'). I suggest going through the manuscript and replacing these instances by more specific terms. On l.54 e.g. This focus on meteorological drought? Another example is: 'this issue' on l.92: which issue?

We revised the text to specify these instances.

• L. 127-132: I would clearly highlight that this statement is an assumption because it is not intuitive and also not in line with some of the literature out there. I agree that this assumption is in some cases useful, I just think it should be openly declared as an assumption and not as a fact.

We reworded to clarify.

- L. 157: what about the non-member states such as Norway or Switzerland? They are still part of the analysis.

We reworded to clarify that EU neighbor countries are also included in the platform.

- L. 206-208: I think that this description of the Lomax function is not entirely correct. The Lomax function has 0 support and a mean of kappa/(alpha-1) and is only defined for alpha > 1. The statement that the 'location parameter is equal to zero' is therefore wrong. Please also provide a reference to the publication, where this distribution was first introduced.

We reworded the sentence in order to avoid confusion. Here we just stated that the Lomax can be derived from the GPD if the μ parameter of the GPD (location parameter) is assumed to be equal to 0. We removed these details since it is not relevant for the paper.

- Would move l. 217-218 to l.211. Was this goodness-of-fit testing done for the same catchments used in this study (or a subset of them)?

This test was done for all the cells in the European domain, as well as at global scale. To better detail the reliability of the methodology for drought analysis the entire paragraph has been reshaped, and it now better describes the validation at global and European scale performed during the implementation of the indicator for operational drought monitoring. We also added a link to the reports that use the indicator for monitoring recent drought events in Europe.

- L. 289: It seems that there are just two river basins in Denmark which were actually considered in this analysis. Are they representative of the Danish hydrology? I would maybe refrain from specially mentioning it.

As stated in the methodology, only the river basin with a drainage area greater than 1000 km2 were analyzed. We rephrased to highlight that this results refers only to the main rivers in Denmark.

- Were livestock and domestic use kept constant in future? If so, why? If not, please shortly describe the estimation procedure used.

Livestock use remains constant, since no future projections were available. However, we expect a limited impact of this assumption on the main conclusions of our work due to the relatively low water use volumes in this sector throughout most of Europe. Work is ongoing to improve this aspect in future applications.

We overall reshaped the water use description to make clearer the adopted procedure for the different sectors. This is done by separating the description of the downscaling procedures used to obtain high-resolution maps from national-level data, from the approaches adopted for the projection of water uses, including the assumption made for livestock.

- L. 246: I agree with reviewer 2 that some significance testing would be highly desirable. If this is not done, please at least mention that no significance testing was done and that the definition of 'robustness' entirely relies on the sign but not the significance of change (l. 243- 246).

We have reformulated the sentence to further underline that a robustness test was performed based only on the agreement on the sign of the changes, and that it refers to the robustness of the ensemble mean.

- L. 442: I think this decrease in summer streamflow is not only due to less precipitation but also smaller snowmelt contributions see e.g. [*Stahl et al.*, 2016; *Jenicek et al.*, 2018].

We agree. The text has been edited to account for this other cause of streamflow reduction in summer.

• L. 452-454: I think that these statements need references. And as mentioned above, I think that this section about water use should be expanded.

We added a reference to water footprint in Europe. Also, the discussion on the role of water use, as well as its uncertainty, has been extended.

• L. 477-482: I think that this information is partially redundant and could be merged with the introduction and methods.

We agree that this paragraph sounds a little repetitive. This paragraph was meant to reiterate some of the key messages of the study as part of the summary part of the section. We reworded and shrank the paragraph to better differentiate from the introduction.

• Data availability: please specify from whom the data can be requested and what subset will be made available through the JRC data hub.

All the data produced by the JRC are freely available upon request. We better clarify this information. Details on the data distribution through the DRKMC data hub are still under definition, but we clarified that all the main outputs will be distributed but not the whole dataset, since the research produced TBs of data.

**Figures**

• I would add lables (a), (b),… to all subfigures presented. This would facilitate referencing in the text.

After re-reading the full text, we had the impression that the figures and sub-figures were overall clearly referenced even without labels. Hence, we prefer to leave the figure as they are, since some of them are already quite dense.

• Figure 2: When I first looked at the figure, I was confused by the two inversed scales. I understand now why they are useful. Still, a note in the figure caption would be helpful.

We added a note to the figure caption.

• Figure 3: I would indicate the medians for all PDFs to facilitate following section 3.1.3. The tick marks are clearly not 'uniformly' spaced as indicated in the figure caption.

As discussed in the text, the median isn't the only interesting feature to analyze in these plots, since the spread of the plot is quite important as well. Overall, we think that adding the median will only overcrowd an already quite busy plot. Also, changes in the median values can be observed in Fig. 4.

• Figure 4: I would indicate that the x-axis labels differ between subplots.

We added a note in the caption.

• Figure 5: I would indicate the country abbreviations on Figure 1d and I would limit the scale to +/- 4 in order to improve legibility and reduce white space.

• Figure 6: Similarly, I would limit these figures to +/- 3.

The limits of the x-axes have been adjusted to better fit the data.

• Tables 1 and 2: I would write 'total' instead of 'tot'

Done.

**Suggested edits**

- L.23: suggest rephrasing to 'is expected to experience' instead of 'sees' to be less deterministic.

Done.

- I suggest replacing the keyword 'low-flow index' by 'human water use' and 'frequency analysis'.

We revised the keywords.

- L.44-46: suggest rephrasing the sentence to something like: 'A specific drought type may be perceived most relevant for a given application and various indicators may experience different effects of climate change.'

We reworded the sentence.

- L. 49: suggest replacing 'climate projection of' to 'impact of climate change on'.

Done.

- L. 53: suggest replacing 'with the latter usually requiring' by 'whose analysis usually requires'.

Done.

- L. 66: maybe rather use 'domain' instead of 'extent'?

We reworded the sentence.

- L. 76: remove 'of' in front of 'past'.

Done.

- L. 81: has it already shifted or is it still shifting?

Done.

- L. 88: the word 'annual' confused me here. Was the threshold not determined using daily streamflow values?

We remove the word annual.

- L. 90: remove 'a' in front of 'key'.

Done.

- L. 92: By 'this issue', do you mean 'future drought changes under the influence of climate change and water abstraction'?

We reworded the sentence.

- L. 95: would rephrase to: ' the threshold level method for event extraction, which allows for a detailed frequency analysis of different streamflow drought characteristics including severity, duration, and frequency.'

We reworded this sentence according to your suggestion.

- L. 100: incorporate's'

Done.

- Would remove l.105-108 or merge with l.56-58 to avoid redundancy.

We agree that merging the two improved the readability.

- L. 112-113: What is the purpose of this sentence in the introduction? Would more this to the methods section.

We removed the sentence, since this is discussed more in depth in the methods section.

- L. 124: suggest replacing 'over' by 'compared to'.

Done.

- L. 126: what does 'on average' refer to? All model runs? Also suggest replacing 'at' by 'in'.

Yes, we clarified that in the text.

- L. 134: suggest adding 'a' in front of '5'.

Done.

- L. 140: snow accumulation and melt.

Done.

- L. 161: I wonder how relevant 'paddy-rice irrigation' is in Europe? If this is irrelevant in Europe, I would exclude its description (l. 164-165).

Rice production is relevant in southern Europe.

- L. 166: is 'a' function of…

Done.

- L. 172: what about non-urban but still populated areas?

This sentence is meant to only highlight that commercial/service water use is considered negligible. A full description on how the downscaling was performed is clearly out of the scope of this paper. It can be found in the referenced publication, including how areas with sparse population and touristic population is accounted.

- L. 173: corresponding to what?

We clarified that in the new version of the manuscript.

- L.185: please mention that Q85 refers to exceedance probabilities.

Done.

- L. 190: 'volume' instead of 'area'?

You are correct that this is a volume of water. However, here we refer to the fact that in a temporal plot Q-t the deficit is represented by the area between the discharge timeseries and the threshold.

- L. 192: suggest removing 'temporal'.

Done.

- L. 194-198: I would reorganize this sentence and swap 2) with 1) because of the order of the two elements in the first part of the sentence.

Done.

- L. 198-201: This information seems to be redundant with information provided on l.194-195 and can in my opinion be removed.

Agree.

- L. 202: Following this 'drought' definition,..

Done.

- L. 203: 'was' derived. Would also remove 'huge'.

Done.

- L. 215-217: what is the purpose of this sentence here?

This sentence refers to a paper where an implementation and validation of the same drought method has been performed over Europe. This section has been further expanded to highlight the previous studies on the validation of Lisflood for drought modelling.

- L. 221-223: I think this sentence needs rephrasing.

We reworded this paragraph.

- L. 231-234: rephrasing recommended.

We reworded this paragraph to simplify the description.

- L. 237: aggregate'd'.

Done.

- L. 251-255: I think this sentence needs rephrasing.

The full paragraph has been reworded.

- L. 260: Atlantic 'region'.

Done.

- L. 266: replace 'halve' by 'half'.

Done.

- L. 287: the rest 'of the region' shows…

Done.

- L. 297: longer droughts 'with increasing GWL'.

Done.

- L. 300: longer 'than 5 days'. Applies to the whole section: longer than 'what'?

The whole section discusses "changes from the reference 1981-2010". We clarified where needed.

- L. 307: last longer 'than in the reference period'.

This sentence has been reworded.

- L. 318-319: This statement does not seem to be true for the Boreal region at 3K.

Here changes of 15 days or more are discussed, which correspond to the dark blue shade.

- L. 324: Figure three seems to show densities not distributions.

Done.

- L. 357: clear'ly'

Done.

- L. 361: maybe specify, i.e. the rarer events.

Done.

- L. 403: 'change' patterns?

Done.

- L. 405-408: think this sentence needs rephrasing.

We reworded the sentence.

- L. 409: would replace 'temporal horizons' with 'time periods instead of GWL'.

Done.

- L. 414 and other instances: I would not use the word 'negative' and 'positive' because they don't seem to be used in an objective sense in terms of – and + but rather in the sense of perception. See also L. 431 'positive', which could be replaced by decreasing.

We reworded the instances where negative/positive is not used as in +/-.

- L. 422: would replace 'symmetrically' by 'in contrast'.

Done.

- L. 425: 'are' instead of 'is'.

Done.

- L. 434: supply '(precipitation)'?

Done.

- L. 443: would delete 'for the Paris warming targets'.

Done.

- L. 447: What does 'this' refer to?

We reworded the whole paragraph to clarify.

- L. 450: would delete 'months'.

Done.

- L. 467: region 'are' expected

Done.

- L. 468: would replace 'symmetrically' by 'in contrast'.

Done.

- L. 473: by 'mostly well defined', do you mean 'robust'?

Done.

- L. 487: would swap the order of 'here' and 'analysed'.

Done.

- L. 490: agricultural 'land' exposed in 'the' southern…

Done.

- L. 492: what does 'this' refer to?

We reworded to clarify.

- L. 494: less 'frequently'.

Done.

**References used in this review**

Graham, N. T. et al. (2018), Water sector assumptions for the Shared Socioeconomic Pathways in an integrated modeling framework, *Water Resour. Res.*, *54*(9), 6423–6440, doi:10.1029/2018WR023452.

Jenicek, M., J. Seibert, and M. Staudinger (2018), Modeling of future changes in seasonal snowpack and impacts on summer low flows in Alpine catchments, *Water Resour. Res.*, *54*(1), 538–556, doi:10.1002/2017WR021648.

Stahl, K., M. Weiler, I. Kohn, D. Freudiger, J. Seibert, M. Vis, and K. Gerlinger (2016), *The snow and glacier melt components of streamflow of the river Rhine and its tributaries considering the influence of climate change*, Freiburg.

[revised manuscript text omitted]